# The Devil is in the Spectrum: Mitigating Representation Collapse in LLMs via Topologically Regularized Side-Path

Yiheng Tao [* 1 2]   Kaiwen Cheng [* 1 3]   Yao Lu [✉ 2]   Chang Liu [✉ 4]   Jie Chen [✉ 5 2 1 3]

## Abstract

Large Language Models (LLMs) are fundamentally limited by representation collapse, a bottleneck that severely degrades long-context performance. We identify that existing approaches risk drifting into one of two pathological extremes: homogenization collapse (e.g., attention sinks causing rank deficiency) and isolation collapse (e.g., local attention causing context disconnection). Through spectral analysis of attention dynamics, we derive an intrinsic trade-off between mixing efficiency (spectral gap) and information capacity (effective rank) that standard mechanisms struggle to balance. To resolve this dilemma, we propose the Topologically Regularized Side-Path (TRSP), a non-invasive architectural intervention that achieves spectral balance. TRSP employs a parameter-free Triangular Box mechanism, scaled by a lightweight, length-aware gate, to regularize the token interaction topology. By integrating proximal coupling to preserve effective rank and distal propagation to support non-degenerate mixing, TRSP promotes a geometrically healthier transition operator without altering core attention. Experiments show significant improvements across general capabilities and long-context benchmarks. Notably, on NoLiMa at $8\times$ the training length, TRSP retains $83\%$ accuracy and surpasses the Differential Transformer and Gated Attention by approximately 30 and 50 percentage points, respectively. Code available at: https://github.com/Eziotao-tyd/TRSP.

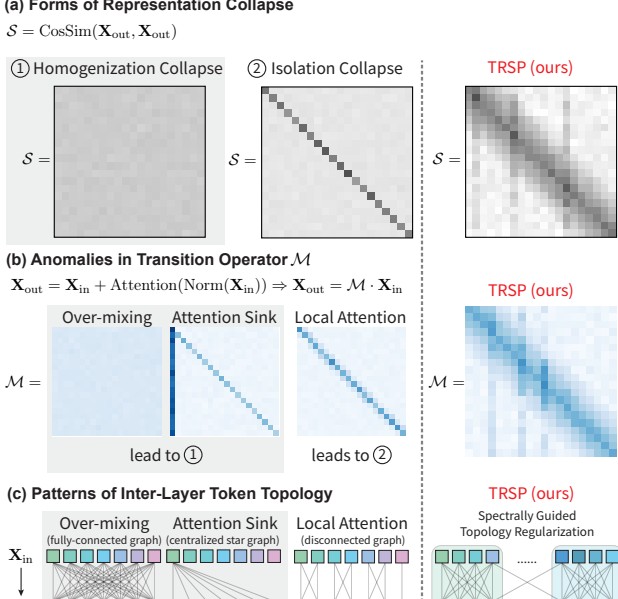

**(a) Forms of Representation Collapse**
$\mathcal{S} = \mathrm{CosSim}(\mathbf{X}_{\mathrm{out}}, \mathbf{X}_{\mathrm{out}})$

**(b) Anomalies in Transition Operator** $\mathcal{M}$
$\mathbf{X}_{\mathrm{out}} = \mathbf{X}_{\mathrm{in}} + \mathrm{Attention}(\mathrm{Norm}(\mathbf{X}_{\mathrm{in}})) \Rightarrow \mathbf{X}_{\mathrm{out}} = \mathcal{M} \cdot \mathbf{X}_{\mathrm{in}}$

**(c) Patterns of Inter-Layer Token Topology**

*Figure 1.* **Analysis of representation collapse.** (a) Token Similarity $\mathcal{S}$. Attention may degenerate into homogenization collapse (indistinguishable tokens) or isolation collapse (context failure). (b) Transition Operator $\mathcal{M}$. Homogenization may stem from over-mixing or an attention sink, while isolation arises from local attention. (c) Topological Connectivity. These pathologies originate from fully connected/centralized star graphs versus disconnected graphs, respectively. By applying spectrally guided topology regularization, our approach constructs a non-degenerate transition operator $\mathcal{M}$ that effectively mitigates representation collapse.

## 1. Introduction

Large Language Models (LLMs) achieve strong natural language understanding and generation, yet they suffer from a fundamental pathology of representation collapse, identified in studies of over-mixing and rank collapse (Dong et al., 2021; Noci et al., 2022; Wu et al., 2024; Barbero et al., 2024; Saada et al., 2025) by analyzing how information propagates through transformers: as the context length increases (in *width*), repeated mixing by the attention mechanism (in *depth*) drives token representations toward a low-dimensional, uninformative space.

In parallel with these theoretical findings, empirical research has explored countermeasures. First, the ubiquitous

[*]Equal contribution   [1]School of Electronic and Computer Engineering, Peking University, Shenzhen, China [2]Pengcheng Laboratory, Shenzhen, China [3]AI for Science (AI4S)-Preferred Program, Peking University Shenzhen Graduate School, China [4]Department of Automation and BNRist, Tsinghua University, Beijing, China [5]School of Intelligence Science and Engineering, Harbin Institute of Technology, Shenzhen, China. Correspondence to: Yao Lu <yaolubrain@gmail.com>, Chang Liu <liuchang2022@tsinghua.edu.cn>, Jie Chen <chenj@pcl.ac.cn>.

*Proceedings of the 43rd International Conference on Machine Learning*, Seoul, South Korea. PMLR 306, 2026. Copyright 2026 by the author(s).

*Figure 2.* **Achieving Spectral Balance via Topological Regularization.** (a) Spectral Trade-off. Analysis of $\mathcal{M}$ reveals an intrinsic trade-off between Information Capacity (measured by effective rank, $R_{\text{eff}}$) and Mixing Efficiency (measured by spectral gap, $\gamma$). The patterns in Fig. 1 collapse into either homogenization (high $\gamma$, low $R_{\text{eff}}$) or isolation (low $\gamma$, high $R_{\text{eff}}$). We require spectral balance to resolve this dilemma. (b) We introduce the Topologically Regularized Side-Path (TRSP) as a non-invasive regularizer. It induces a topology in which proximal interactions in shallow layers preserve feature distinctiveness (Rank-Preserving), while distal propagation in deep layers supports global mixing (Gap-Preserving).

attention sink phenomenon (Xiao et al., 2024; Gu et al., 2025)—in which massive attention weight is allocated to initial or special tokens—has been interpreted as a natural mechanism to arrest the aforementioned over-mixing (Barbero et al., 2025). Second, architectural designs based on local or sliding-window attention (Jiang et al., 2023; Team et al., 2024; Beltagy et al., 2020; Child, 2019) restrict the per-layer receptive field to reduce the cost of long-context attention, thereby limiting the immediate scope of token mixing. We argue that both countermeasures avoid one failure mode at the cost of another. Under a unified spectral lens, we categorize these phenomena as two pathological extremes of representation collapse: (1) homogenization collapse (e.g., over-mixing or the attention sink), characterized by critically low effective rank (Wu et al., 2024); and (2) isolation collapse, characterized by critically low contextual coherence (Ethayarajh, 2019), in which mixing fails to bridge distant tokens (e.g., truncated local attention).

We visualize these pathologies in Figure 1. First, the token similarity matrices in (a) show that tokens either degenerate into indistinguishable noise (homogenization) or fail to capture context (isolation). Second, we trace this to the transition operator $\mathcal{M}$ in (b), where $\mathbf{X}_{\text{out}} = \mathcal{M} \cdot \mathbf{X}_{\text{in}}$. The homogenization pattern appears as a dense or sink-dominated operator, whereas isolation corresponds to a banded operator. Both patterns fail to support effective information flow. Third, inter-layer topology analysis in (c) links these behaviors to token connectivity: homogenization stems from fully connected or centralized star graphs, whereas isolation stems from disconnected graphs. This raises a more fundamental question: what spectral conditions characterize a non-degenerate operator $\mathcal{M}$, and why do standard mechanisms fail to satisfy them simultaneously?

To address this, we analyze the spectral properties of the row-normalized operator $\mathcal{M}$ via singular value decomposition, as shown in Figure 2 (a). We use two metrics whose maximization is desirable: effective rank ($R_{\text{eff}} = \sum \sigma_i^2$) (Rudel-

son & Vershynin, 2007; Roy & Vetterli, 2007), which measures information capacity via the heaviness of the singular value distribution's tail, and spectral gap ($\gamma = 1 - \sigma_2$), which measures mixing efficiency via the separation of the leading singular values.[1] We derive an intrinsic relationship between them: $R_{\text{eff}} = 1 + (1 - \gamma)^2 + \sum_{i=3}^{N} \sigma_i^2$. This reveals a fundamental spectral trade-off: maximizing $\gamma$ inevitably suppresses $R_{\text{eff}}$ unless the tail ($\sum_{i=3}^{N} \sigma_i^2$) is explicitly preserved. Empirically, models are often trapped in this dilemma: homogenization (high $\gamma$, low $R_{\text{eff}}$), where strong mixing collapses the manifold, or isolation (low $\gamma$, high $R_{\text{eff}}$), where weak mixing preserves dimensionality but halts propagation.

To resolve this trade-off and target spectral balance (i.e., jointly improving $R_{\text{eff}}$ and $\gamma$ without collapsing to either extreme), we propose the Topologically Regularized Side-Path (TRSP). We retain the standard attention computation and augment the transition operator with an additive side-path. This design injects topological regularization without disrupting the core attention mechanism. TRSP enforces a hierarchy that combines proximal coupling (to preserve the tail and $R_{\text{eff}}$) with distal shortcuts (to support high $\gamma$). Specifically, TRSP introduces two key components: (1) the Triangular Box ("triBox") mechanism, a parameter-free operator implemented via cascaded causal box filters with exponential bandwidth expansion. It creates a seamless scale transition from proximal interactions to distal connections across layers. (2) A long-context gate, a lightweight gain controller. With only $\approx 50$ learnable parameters, the gate regulates side-path strength from context length and layer depth via the coverage ratio, calibrating triBox injection across scales.

We conducted extensive experiments evaluating general capabilities in post-training scenarios (MMLU (Hendrycks

---

[1]For notational simplicity, throughout this discussion $\sigma_i$ denotes the singular values normalized by the largest singular value of $\mathcal{M}$; hence $\sigma_1 = 1$.

et al., 2021), HellaSwag (Zellers et al., 2019)) and long-context extrapolation in both pre- and post-training settings (RULER (Hsieh et al., 2024), NoLiMa (Modarressi et al.)). Our results confirm that TRSP effectively corrects spectral anomalies and consistently outperforms strong baselines. Notably, on NoLiMa, TRSP retains 83% accuracy at $8\times$ the training length, surpassing the Differential Transformer (Ye et al., 2025) and Gated Attention (Qiu et al., 2026) by approximately 30 and 50 percentage points, respectively.

## 2. Related Work

### 2.1. Representation Collapse and Spectral Analysis

Prior work has analyzed Transformer representation collapse along two complementary axes: layer depth and context length. *Along the depth axis,* Dong et al. (2021) show that pure attention loses rank doubly exponentially with depth, driving representations toward a rank-1 subspace. This pathology has been linked to vanishing gradients, signal-propagation failure, and representation degeneration (Noci et al., 2022; He et al., 2023; Wu et al., 2024; Geshkovski et al., 2025; Saada et al., 2025; Gao et al., 2019). This phenomenon parallels over-smoothing in Graph Neural Networks (Keriven, 2022; Wu et al., 2023) and has been explicitly formalized within Transformers, where self-attention is shown to behave as a low-pass filter that homogenizes features across layers (Shi et al., 2022; Wang et al., 2022; Nguyen et al., 2023). A complementary perspective shows that whether such smoothing is unavoidable depends on the eigenspectrum of the value and projection weights, leaving room for structural intervention (Dovonon et al., 2024). *Along the length axis,* Veličković et al. (2025) show that, under bounded-logit assumptions, increasing context length can drive softmax attention toward uniform mixing, while Barbero et al. (2024) identify an over-squashing pathology in causal architectures, in which the unidirectional information flow renders the final-token representations of distinct sequences arbitrarily close. These studies primarily diagnose collapse via spectral and geometric tools (Roy & Vetterli, 2007; Pennington et al., 2017; Ethayarajh, 2019), or attempt to mitigate it through targeted modifications of attention masks, normalization, or weights. We instead (i) unify these homogenization phenomena (both depth- and length-wise) with the opposite extreme of isolation collapse as two ends of a single trade-off between effective rank and spectral gap, and (ii) propose an explicit topological regularizer to balance both simultaneously.

### 2.2. Long-Context Modeling and Attention Sink

Long-context modeling remains a central concern for LLMs, as models often exhibit position-dependent context utilization, performing worse on information located in the middle of long sequences (Liu et al., 2024b).

*Architectural and computational approaches.* Early work explored recurrence (Dai et al., 2019) or sparse attention patterns, including the Sparse Transformer (Child, 2019), Longformer (Beltagy et al., 2020), and BigBird (Zaheer et al., 2020). Recent advances focus on efficient computation, including FlashAttention and its successors (Dao et al., 2022; Shah et al., 2024) and Ring Attention (Liu et al., 2024a), alongside KV cache compression techniques (Zhang et al., 2023; Ge et al., 2024; Wu & Tu, 2024). Positional encodings and extension strategies such as ALiBi (Press et al., 2022), RoPE (Su et al., 2024), and YaRN (Peng et al., 2024) facilitate length extrapolation but are largely orthogonal to the spectral degradation we focus on. Beyond softmax-attention Transformers, alternative architectures such as state-space models (Gu & Dao, 2024) and recurrence-based linear models (Peng et al., 2023) replace attention with subquadratic primitives; in this work, we focus on improving the spectral behavior of the standard softmax-attention Transformer rather than replacing it.

*Attention sink and attention modifications.* The attention sink phenomenon (Xiao et al., 2024), in which massive weight is allocated to the initial token, has prompted competing interpretations. One line of work views sinks as pathological artifacts tied to massive activations (Sun et al., 2024), activation outliers (Kaul et al., 2025), or softmax-induced first-token bias (Gu et al., 2025), and proposes mitigations via softmax reformulations such as sigmoid attention (Ramapuram et al., 2025) and softmax-1 (Kaul et al., 2025), or via gating mechanisms (Bondarenko et al., 2023). A second line actively leverages sinks for streaming inference (Han et al., 2024; Xiao et al., 2024). Barbero et al. (2025) reinterpret sinks as a learned mechanism by which deep Transformers arrest over-mixing. We argue that sinks replace one homogenization mode (uniform over-mixing) with another (first-token concentration), both characterized by low effective rank. Closely related to our work, recent attention modifications—Differential Transformer (Ye et al., 2025), which cancels attention noise via the difference of two softmax maps, and Gated Attention (Qiu et al., 2026), which applies a query-dependent sigmoid gate after the SDPA output to eliminate sinks—reshape the attention computation itself. In contrast, TRSP introduces a non-invasive side-path that targets the spectral structure of the transition operator $\mathcal{M}$ without altering standard attention, making it complementary to these methods and readily composable with existing architectures.

## 3. Methodology

### 3.1. Overview

As analyzed in §1, we introduce the Topologically Regularized Side-Path (TRSP), a lightweight branch added in parallel to the standard attention layer. As illustrated in

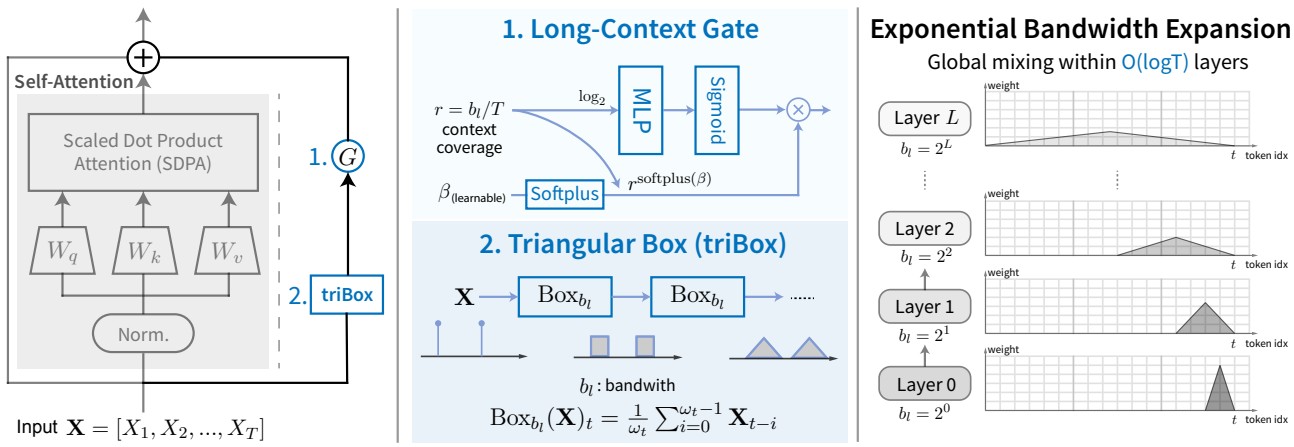

*Figure 3.* **Overview of TRSP.** Left: The overall architecture showing TRSP as a parallel branch. Middle: Detailed implementations of the Long-Context Gate (top) and the triBox mechanism (bottom). Right: The hierarchical connectivity pattern showing how bandwidths expand exponentially ($2^\ell$) across layers.

Figure 3, TRSP applies a causal triangular filter with a layer-dependent bandwidth to the hidden states and adds the gated output back to the residual stream, leaving the attention computation unchanged. The triBox branch provides a fixed multi-scale route for long-range signal propagation, and the long-context gate controls its strength based on context coverage. This side-path serves as a non-invasive structural bias toward spectral balance.

### 3.2. The Triangular Box (triBox) Mechanism

We design triBox to regularize the layer-wise mixing operator $\mathcal{M}$ towards spectral balance: sufficient mixing to avoid isolation and sufficient capacity to avoid homogenization. Concretely, we target a non-vanishing spectral gap while preventing the singular spectrum of $\mathcal{M}$ from collapsing to rank one. Below, we relate this goal to the two collapse modes in §1 via two quantities.

**Mixing efficiency (spectral gap).** For a row-normalized transition operator, the spectral gap $\gamma(\mathcal{M}) = 1 - \sigma_2$ quantifies how quickly non-stationary components contract, thereby indicating the strength of global token mixing (Levin & Peres, 2017). A vanishing gap indicates weak long-range mixing and aligns with isolation collapse. In theory, we study $\gamma$ on $\mathcal{M}$; in experiments, we report the Signal Propagation Rate (SPR) as the perturbation gain $\|\delta_{\text{out}}\|/\|\delta_{\text{in}}\|$, used as an empirical proxy for propagation strength, following Transformer signal-propagation studies (Noci et al., 2022; Saada et al., 2025) and sensitivity viewpoints (Gouk et al., 2021).

**Information capacity (effective rank).** Even with adequate mixing, representations can still collapse if energy concentrates on a few singular directions—the hallmark of homogenization collapse. We therefore monitor the spread of the singular spectrum of $\mathcal{M}$. Following (Roy & Vetterli,

2007), we use effective rank as a continuous notion of dimensionality; in this paper we quantify it by the stable rank $R_{\text{stab}}(\mathcal{M}) = \|\mathcal{M}\|_F^2/\|\mathcal{M}\|_2^2$ (Cohen et al., 2015), which coincides with $R_{\text{eff}} = \sum_i \sigma_i^2$ in §1 under singular value normalization $\sigma_1 = 1$.

#### 3.2.1. IMPLEMENTATION VIA CASCADED BOX FILTERS

triBox is a channel-wise causal triangular convolution implemented by cascading two causal box filters (moving averages), rather than via dense matrix multiplication. Let $\mathbf{X} \in \mathbb{R}^{T \times d}$ denote the token hidden states and let $b \in \mathbb{N}$ be the box window length (i.e., bandwidth). For each channel independently, the causal box filter computes a length-normalized moving average over the past $b$ positions, using only the $t+1$ tokens available at time $t$ when $t < b$:

$$\text{Box}_b(\mathbf{X})_t = \frac{1}{w_t} \sum_{i=0}^{w_t-1} \mathbf{X}_{t-i}, \qquad (1)$$

where $w_t = \min\{b, t+1\}$ is the effective window size. A naive sliding window costs $O(Tb)$; we instead maintain prefix sums $\mathbf{S}_0 = \mathbf{0}$ and $\mathbf{S}_{t+1} = \mathbf{S}_t + \mathbf{X}_t$. With start index $a_t = \max\{0, t-b+1\}$, the same filter is evaluated in $O(1)$ per token as

$$\text{Box}_b(\mathbf{X})_t = \frac{\mathbf{S}_{t+1} - \mathbf{S}_{a_t}}{t - a_t + 1}, \qquad (2)$$

which is algebraically identical to Eq. (1) and costs $O(Td)$ for any $b$. triBox applies the box filter twice:

$$\text{triBox}_b(\mathbf{X}) = \text{Box}_b\big(\text{Box}_b(\mathbf{X})\big). \qquad (3)$$

When $t \geq 2b-2$ both box windows are full, the cascade is equivalent to a single causal triangular convolution,

$$\text{triBox}_b(\mathbf{X})_t = \sum_{r=0}^{2b-2} h_b(r)\,\mathbf{X}_{t-r}, \qquad (4)$$

with weights $h_b(r) = (b - |r - (b-1)|)/b^2$ that decay linearly toward the past and sum to one on $\{0, \ldots, 2b-2\}$, yielding a smoother frequency response than a single rectangular window.

### 3.2.2. DYNAMIC BANDWIDTH EXPANSION

To cover multiple scales without making each layer dense, the box-filter window length grows exponentially with depth. For layer $\ell \in \{0, \ldots, D-1\}$, we set

$$b_\ell = \min\{2^\ell, T\}, \quad (5)$$

where $T$ is the current sequence length and $D$ is the number of layers. At layer $\ell$, triBox is a local triangular mixer with bandwidth $b_\ell$, which limits homogenization within its causal receptive field. Across layers, the dyadic schedule $b_\ell = 2^\ell$ superposes connections at scales $\{\pm 2^\ell\}$, inducing a sparse Cayley graph on $\mathbb{Z}_T$ (Figure 3, Right). Shallow layers therefore realize proximal coupling (preserving $R_{\text{eff}}$), while the stacked topology provides long-range shortcuts that support global mixing (bounding $\gamma$), matching the division in Figure 2.

### 3.2.3. SPECTRAL PROPERTIES

We now state the rank and gap guarantees that the triBox topology confers on the composite operator $\mathcal{M}$; full proofs are deferred to Appendix C, and their link to the model's inference-error bound to Appendix D.

**Rank lower bound.** Standard attention can concentrate energy on a single sink token, approaching a rank-one matrix. The triangular kernel instead spreads energy across its window, so the Frobenius energy of the triBox operator grows linearly with length, $\|M_{\text{tri}}\|_F^2 = \Theta(T)$. This scaling lower-bounds the effective rank of $\mathcal{M}$ away from one even under sink collapse (Appendix C.4), so the side-path preserves usable dimensionality as $T$ grows.

**Non-degenerate mixing.** Superposing the dyadic bandwidths across layers wires a cyclic Cayley graph on $\mathbb{Z}_T$, whose algebraic connectivity—the unnormalized spectral gap $\mu_2$, i.e., the smallest non-zero Laplacian eigenvalue—is constant in $T$ (Appendix C.3). This secures robust absolute energy flow, while the corresponding normalized gap decays only as $\Theta(1/\log T)$. A dense variant offsets this decay through depth but costs $O(Td \log T)$ per layer, so we keep the sparse design, retaining the constant $\mu_2$ at $O(Td)$ complexity.

### 3.3. The Long-Context Gate

triBox fixes the side-path topology and per-layer mixing geometry (§3.2). A fixed scalar injection into the residual stream is brittle: shallow layers with small $b_\ell$ can oversmooth short contexts, and different $(T, \ell)$ pairs require different side-path gains relative to attention (replacing $g_\ell$ with a static scalar reduces MMLU to 21.16%; §5). We introduce the long-context gate, a global, input-agnostic gain $g_\ell$ that depends only on the coverage ratio $r_\ell = b_\ell/T$.

### 3.3.1. FORMULATION

The $\ell$-th layer update is

$$\mathbf{X}_{\text{out}} = \mathbf{X}_{\text{in}} + \text{Attn}(\mathbf{X}_{\text{in}}) + g_\ell \, \text{triBox}_{b_\ell}(\mathbf{X}_{\text{in}}), \quad (6)$$

where $g_\ell \in (0, 1)$ scales the side-path branch. The gate depends on the coverage ratio $r_\ell = b_\ell/T$, i.e., the fraction of the sequence spanned by the local triBox window at layer $\ell$ (Figure 3, Middle). With learnable decay exponent $\phi = \text{softplus}(\beta) > 0$, we define

$$g_\ell = \sigma\big(\text{MLP}(\log_2 r_\ell)\big) \cdot r_\ell^\phi. \quad (7)$$

Here $\sigma(\cdot)$ is the sigmoid and $\text{MLP}(\cdot)$ is a small Multi-Layer Perceptron; we feed $\log_2 r_\ell$ so that dyadic changes in coverage map to approximately linear inputs for the MLP. The only trainable gate parameters are the MLP weights and the scalar $\beta$ (shared globally across layers and tokens; $\approx 50$ parameters in our setups). Unlike input-dependent gates in Gated Attention (Qiu et al., 2026), $g_\ell$ does not depend on hidden states; it calibrates injection from $(T, \ell)$ alone.

### 3.3.2. INTERACTION WITH SPECTRAL BALANCE

Scaling by $g_\ell$ sets the effective weight of the triBox mixing matrix in the composite layer operator $\mathcal{M} = I + A_{\text{attn}} + g_\ell M_{\text{tri}}^{(\ell)}$. The triBox branch itself carries the rank and gap guarantees; the gate specifies *how strongly* that structural component enters the composite operator as $T$ and $\ell$ vary.

**Input-agnostic structural gain.** $g_\ell$ depends only on $(T, \ell)$ via $r_\ell = b_\ell/T$. The side-path injection therefore cannot be suppressed by input-specific activation patterns, attention sinks on particular tokens, or adversarial perturbations of $\mathbf{X}$. Instead, $g_\ell$ is fixed for a given forward pass once the sequence length and layer index are known; it acts as a *structural* gain schedule rather than a content-dependent switch. This isolates a predictable contribution from the topologically regularized branch within $\mathcal{M}$.

**Asymptotic stability.** The factor $r_\ell^\phi$ captures how side-path gain should scale as $T \to \infty$: because $r_\ell = b_\ell/T$ shrinks with length, a learned $\phi > 0$ increases the relative triBox contribution and counters dilution of the fixed topology. In practice, this provides a simple, length-aware calibration rule for how strongly $M_{\text{tri}}^{(\ell)}$ enters the composite operator at each layer.

**Remark.** It is important to clarify the scope of these claims. While the proposed topology strictly ensures the spectral gap and effective rank of the TRSP residual branch itself,

| Method | Extra Params | MMLU | HellaSwag | | Spectral Diagnostics (Avg.) | | | |
| --- | --- | --- | --- | --- | --- | --- | --- | --- |
| | | Acc ↑ | Acc ↑ | PPL ↓ | SPR ↑ | Rank ↑ | Anisotropy ↓ | Flatness ↑ |
| Llama-3.2-1B (Raw) | - | 35.09 | 25.94 | **1.67** | **1.60** | 1.40 | 0.21 | 0.42 |
| LoRA | 5.6M | 36.01 | 27.97 | 3.58 | 1.51 | 1.36 | 0.22 | 0.42 |
| LoRA + Gated Attention (Qiu et al., 2026) | 5.6M + 67.2M | 23.19 | 24.79 | 24.94 | 1.59 | 1.44 | 0.19 | 0.44 |
| **LoRA + TRSP** (Ours) | 5.6M + 50 | **37.76** | **29.26** | 3.05 | 1.59 | **1.93** | **0.18** | **0.69** |

*Table 1.* **Main Results on General Capabilities.** We report additional trainable parameters, MMLU and HellaSwag accuracy (%), HellaSwag perplexity (PPL), and empirical spectral diagnostics (Rank denotes Stable Rank; definitions in §4.1) averaged across layers and datasets. LoRA + TRSP achieves the highest accuracies with only 50 additional trainable parameters beyond LoRA.

the spectrum of the final composite operator is subject to interaction with the data-dependent attention matrix. Through additive perturbation theory, injecting a full-rank component establishes rigorous worst-case spectral guarantees for the entire composite network, as detailed in Appendix C.

## 4. Experiments

We organize the evaluation around four questions: (i) whether TRSP improves standard performance without disrupting the base model, (ii) whether the gains persist when evaluation contexts extend beyond the training window, (iii) whether task-level improvements are accompanied by less-collapsed spectral diagnostics, and (iv) which components are responsible for the observed behavior. We study two complementary settings: post-training on Llama-3.2-1B-Instruct (Dubey et al., 2024) and training a 109M Llama-style transformer from scratch on NoLiMa. In the post-training setting, we compare against the raw model, LoRA (Hu et al., 2022), and Gated Attention (Qiu et al., 2026); in the from-scratch setting, we also compare against the Differential Transformer (Ye et al., 2025). Detailed configurations are provided in Appendix A.

### 4.1. Experimental Setup

**Post-training setting.** For general capability evaluation, hyperparameter sensitivity, and ablations, we fine-tune Llama-3.2-1B-Instruct on Alpagasus-5k (Chen et al., 2024) and evaluate on MMLU (Hendrycks et al., 2021) and HellaSwag (Zellers et al., 2019). For long-context extrapolation, we fine-tune on RULER (Hsieh et al., 2024) with a 4K context window and evaluate at 8K, 16K, and 32K. All post-training variants use the same tuning data and budget where applicable; TRSP is added as a side-path plugin to the LoRA setting.

**From-scratch setting.** To test architectural effects without relying on instruction-tuned priors, we train a 109M-parameter transformer from scratch on NoLiMa (Modarressi et al.) with a 1K context window and evaluate extrapolation up to 8K. NoLiMa requires latent associative reasoning without literal surface overlap between queries and targets,

making it a controlled stress test of long-range information retention.

**Metrics.** Beyond task accuracy, we monitor four empirical spectral diagnostics, computed per layer on the token representations $H_\ell \in \mathbb{R}^{T \times d}$, with two probing each axis of the trade-off in §1. For mixing efficiency, *(i) Signal Propagation Rate (SPR)* is the perturbation gain $\|\delta_{\text{out}}\|/\|\delta_{\text{in}}\|$ obtained by injecting a small Gaussian perturbation at the input; it gauges how strongly a signal propagates rather than being damped (higher is better), following Transformer signal-propagation (Noci et al., 2022; Saada et al., 2025) and Lipschitz-sensitivity (Gouk et al., 2021) analyses. For information capacity, *(ii) Stable Rank* $\|H_\ell\|_F^2/\|H_\ell\|_2^2$ (Roy & Vetterli, 2007; Cohen et al., 2015) estimates the effective dimensionality of the representation (higher is better); a value approaching 1 is the hallmark of homogenization collapse. We complement these with two stability indicators. *(iii) Spectral Flatness*, the ratio of the geometric to the arithmetic mean of the singular values of $H_\ell$ (Gray & Markel, 1974), equals 1 for a perfectly flat, well-conditioned spectrum and tends to 0 as energy concentrates on a few directions; we read it as a proxy for numerical stability, consistent with dynamical-isometry views of well-conditioned learning (Saxe et al., 2014; Pennington et al., 2017). *(iv) Representation Anisotropy*, the average cosine similarity between random token pairs (Ethayarajh, 2019), measures how concentrated the representation cone is (lower is better). We stress that low anisotropy is beneficial *only* when paired with high stable rank—a regime we term *structured isotropy*—because a near-collapsed representation can also appear locally isotropic; we therefore always interpret anisotropy jointly with stable rank.

### 4.2. Main Results: General Capabilities

We first ask whether TRSP improves standard post-training performance while remaining lightweight. Table 1 reports accuracy, perplexity, parameter overhead, and spectral diagnostics after fine-tuning on Alpagasus-5k.

**Performance and Efficiency.** On accuracy metrics, LoRA + TRSP performs best among the post-training methods compared, reaching 37.76% on MMLU and 29.26% on Hel-

| Bench. | Setting | Method | Train | (Extra) Params | 2× | 4× | 8× |
|---|---|---|---|---|---|---|---|
| RULER | Fine-tuning | Llama-3.2-1B (Raw) | 4K | – | 63.85 | 59.23 | 59.55 |
| | | LoRA | 4K | 5.6M | 63.82 | 60.50 | 57.51 |
| | | LoRA + Gated Attention | 4K | 5.6M+67.2M | 45.06 | 45.06 | 40.29 |
| | | **LoRA + TRSP** (Ours) | 4K | 5.6M+50 | **65.68** | **62.78** | **60.38** |
| NoLiMa | From-scratch | Transformer (Base) | 1K | 109.8M | 99.2 | 80.4 | 23.8 |
| | | Transformer + Gated Attention | 1K | 113.5M | 98.4 | 79.3 | 33.6 |
| | | Differential Transformer | 1K | 109.8M | **100.0** | 90.2 | 53.9 |
| | | **Transformer + TRSP** (Ours) | 1K | 109.8M | **100.0** | **98.8** | **83.2** |

*Table 2*. **Long-Context Extrapolation Results.** RULER fine-tunes Llama-3.2-1B-Instruct; NoLiMa trains a 109M Transformer from scratch. Column headers are the ratio $k$ of evaluation to training context length ($k \in \{2, 4, 8\}$): RULER uses a 4K training window and is evaluated at 8K, 16K, and 32K; NoLiMa uses 1K training and is evaluated at 2K, 4K, and 8K. All entries are accuracy (%). TRSP is best among compared methods at $8\times$ in both blocks.

laSwag. This improves on standard LoRA by 1.75 and 1.29 points, respectively, while adding only 50 trainable parameters beyond the LoRA adapters. By contrast, the Gated Attention baseline introduces 67.2M additional parameters in this setup and performs poorly as a post-hoc plugin, suggesting that its benefits may depend on different training dynamics. The raw model retains the lowest HellaSwag perplexity, but TRSP achieves higher downstream accuracy and a lower perplexity than standard LoRA (3.05 vs. 3.58).

**Spectral Diagnostics.** The diagnostic metrics align with the proposed spectral interpretation. Standard LoRA slightly reduces the Stable Rank from 1.40 to 1.36, whereas TRSP increases it to 1.93, indicating a less concentrated representation spectrum. TRSP also recovers most of the SPR reduction introduced by LoRA (1.59 vs. 1.51, close to the raw model's 1.60) and achieves the highest Spectral Flatness (0.69) and the lowest Anisotropy (0.18). These trends support the view that the side-path mitigates representation collapse during post-training.

### 4.3. Main Results: Long-Context Extrapolation

We next test whether TRSP improves performance when evaluation contexts exceed the training window. Table 2 reports both benchmarks under a unified view: RULER under *fine-tuning* and NoLiMa *from scratch*, with columns indexed by the evaluation-to-training length ratio ($2\times$–$8\times$).

**RULER (post-training).** We fine-tune the Llama-3.2-1B-Instruct model on the RULER benchmark with a context length of 4K, then evaluate its performance on extended contexts of 8K, 16K, and 32K ($2\times$, $4\times$, and $8\times$ the training window). As shown in Table 2, standard methods struggle to generalize. Both the raw model and standard LoRA show a clear downward trend as the context length increases. The Gated Attention baseline drops to 40–45% accuracy, suggesting that simply adding a learnable gate fails to learn a generalization law. In contrast, LoRA + TRSP achieves the highest accuracy at every tested length. Crucially, while the trained baselines (LoRA and Gated Attention) degrade

| Method | lr | | |
|---|---|---|---|
| | 6e-4 | 8e-4 | 1e-3 |
| LoRA | 36.01 | 31.87 | 24.47 |
| LoRA + Gated Attention | 23.19 | 24.45 | 24.80 |
| **LoRA + TRSP** (Ours) | **37.76** | **32.10** | **33.60** |

*Table 3*. **Hyperparameter Sensitivity (MMLU Accuracy %).** Comparison of models trained with different learning rates (lr). LoRA + TRSP maintains high performance even at high lrs, whereas baselines degrade or collapse, demonstrating the numerical stability provided by spectral regularization.

faster as the context grows, TRSP sustains the strongest absolute accuracy, indicating that the spectral balance better preserves signal integrity over long sequences. Detailed per-task accuracy for RULER is provided in Appendix B.

**NoLiMa (from scratch).** To isolate the architectural benefits from pre-trained priors, we train models from scratch on the NoLiMa dataset with a fixed 1K context window and test up to 8K. Table 2 compares our method with baselines. The results show that the standard Transformer and Gated Attention variants suffer catastrophic collapse at longer contexts. While the Differential Transformer offers improved robustness at 4K (90.2%), it still degrades significantly to 53.9% at 8K. In comparison, Transformer + TRSP maintains near-perfect performance at 4K and retains a remarkably high accuracy of 83.2% at 8K.

## 5. Ablation and Analysis

This section addresses the remaining experimental questions in §4, in the order they appear below: (iv) which TRSP components matter for post-training accuracy and optimization, and (iii) whether task gains coincide with healthier spectral diagnostics over increasing context. We use Alpagasus-5k / MMLU for component ablations and learning-rate sensitivity, and NoLiMa (from scratch) for length-wise spectral dynamics.

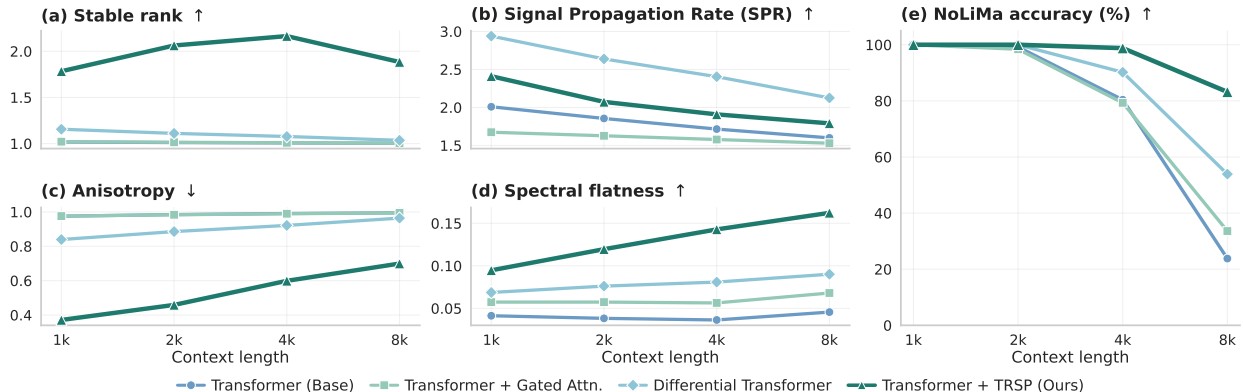

*Figure 4.* **Spectral–performance coupling on NoLiMa** (from scratch, train 1K). **(a–d)** Layer-averaged diagnostics (layers 1–13) vs. evaluation context at $1\times$–$8\times$ training length. TRSP maintains higher stable rank, SPR, and spectral flatness and lower anisotropy than the base Transformer and Gated Attention; the Differential Transformer partially mitigates decay but trails TRSP at long context. **(e)** NoLiMa accuracy (%); values at $2\times$–$8\times$ follow Table 2, with all models at $100\%$ at $1\times$.

| Variant | Extra Params | Acc | Complexity |
|---|---|---|---|
| **TRSP (default)** | 50 | 37.76 | $O(Td)$ |
| w/o long-context gate | 0 | 21.16 | $O(Td)$ |
| Dense (Full-Sweep) | 50 | **38.71** | $O(Td\log T)$ |
| w/o dyadic bandwidth | 50 | 34.84 | $O(Td)$ |
| w/o triangular kernel | 50 | 31.89 | $O(Td)$ |

*Table 4.* **Component ablation (MMLU accuracy, %).** All models fine-tune Llama-3.2-1B with LoRA on Alpagasus-5k. The default TRSP uses $g_\ell$, a sparse dyadic topology, and the triangular kernel. Dense (Full-Sweep) trades a higher per-layer cost for a small accuracy gain.

## 5.1. Hyperparameter Sensitivity: Optimization Stability

Spectral regularization is intended to keep the mixing operator well-conditioned; we therefore test whether TRSP improves robustness to the learning rate in post-training. We fine-tune LoRA, LoRA + Gated Attention, and LoRA + TRSP on Alpagasus-5k and report MMLU accuracy at learning rates $6 \times 10^{-4}$, $8 \times 10^{-4}$, and $1 \times 10^{-3}$ (Table 3).

Standard LoRA is sensitive to this hyperparameter: accuracy falls from 36.01% to 24.47% as the learning rate increases, consistent with unstable updates when the operator lacks structural constraints. Gated Attention remains near 23–25% across all three rates, mirroring its poor post-hoc behavior in §4.2 rather than a length-calibration issue. LoRA + TRSP is substantially more stable at $1\times10^{-3}$ (33.60% vs. 24.47% for LoRA), although its best accuracy still occurs at $6 \times 10^{-4}$ (37.76%). Together with the higher spectral flatness in Table 1, these results are consistent with improved training stability under aggressive optimization.

## 5.2. Ablation Studies: Component Analysis

We ablate the long-context gate $g_\ell$ (§3.3), the dyadic bandwidth schedule $b_\ell = \min\{2^\ell, T\}$ (Eq. (5)), the triangu-

lar triBox kernel, and the sparse-vs-dense topology (Appendix C). All variants fine-tune Llama-3.2-1B with LoRA on Alpagasus-5k; we report MMLU accuracy in Table 4.

**Long-Context Gate.** Replacing the coverage-dependent gate with a single static scalar eliminates all 50 gate parameters but reduces accuracy to 21.16%. A fixed gain cannot match the per-$(T, \ell)$ calibration provided by $r_\ell = b_\ell/T$ and Eq. (7): it either over-injects triBox locally or leaves the side-path too weak to support global mixing.

**Dyadic topology and triangular kernel.** We compare three structural variants. *Dense (Full-Sweep).* Each layer realizes all bandwidths $2^0, \ldots, 2^{D-1}$ in one pass, as in the dense ablation in Appendix C. This achieves 38.71% MMLU ($+0.95$ pt over default) but costs $O(Td\log T)$ per layer; the default TRSP retains 97.5% of that accuracy at $O(Td)$. *w/o dyadic bandwidth.* Fixing $b_\ell = T$ at every layer removes the multi-scale schedule and reduces accuracy to 34.84%. *w/o triangular kernel.* Using a uniform box instead of the cascaded triangular kernel yields 31.89%, indicating that the triangular smoothing is important for limiting local over-mixing relative to a rectangular window.

## 5.3. Spectral Dynamics over Context Length

To connect the NoLiMa extrapolation results in §4.3 to the spectral narrative in §1, we track layer-averaged diagnostics and task accuracy for the from-scratch models as the evaluation context grows from $1\times$ to $8\times$ the 1K training window (Figure 4).

Across this sweep, the base Transformer and Gated Attention show a clear homogenization signature: stable rank and SPR decline, while anisotropy approaches one, indicating that $\mathcal{M}$ loses effective dimensionality and long-range mixing weakens. The Differential Transformer partially slows

this decay—especially for SPR—but does not sustain the same separation at $8\times$. TRSP consistently occupies the more balanced regime targeted in §1: it maintains the highest stable rank and spectral flatness, keeps SPR well above the baselines at long context (consistent with the side-path safety-net view in §3.2), and remains the least anisotropic as length increases.

The same ordering appears in task performance (Figure 4e; Table 2). All models reach near-perfect accuracy at $1\times$, but accuracy diverges sharply at $8\times$, with TRSP retaining 83.2% compared with 23.8% for the base model, 33.6% for Gated Attention, and 53.9% for the Differential Transformer. These context-resolved trajectories show that NoLiMa gains at long evaluation windows co-occur with limiting spectral collapse of $\mathcal{M}$, complementing the improved post-training diagnostics in Table 1 and the trade-off picture in Figure 2.

## 6. Conclusion

We studied long-context degradation by examining the spectral behavior of the transition operator $\mathcal{M}$. Empirically and analytically, standard Transformers tend toward two failure modes: homogenization collapse, in which over-mixing or sink-dominated dynamics reduce the effective rank, and isolation collapse, in which restricted mixing preserves local structure but weakens long-range propagation. Both extremes reduce the usable information dimensionality and degrade performance as the evaluation context grows beyond training.

We proposed TRSP, a non-invasive side-path that regularizes token-interaction topology. By combining the parameter-free triBox operator with a lightweight, length-aware gate, TRSP targets spectral balance: sufficient mixing to avoid isolation while preserving rank and isotropy as context lengthens. Across post-training on Llama-3.2-1B and a 109M from-scratch model, TRSP improves MMLU and HellaSwag, extrapolates more reliably on RULER, and retains 83.2% NoLiMa accuracy at $8\times$ a 1K training window—about 30 and 50 percentage points above the Differential Transformer and Gated Attention, respectively. Context-resolved diagnostics further show that these task gains track trajectories with less spectral collapse of $\mathcal{M}$.

The method adds only $\approx 50$ trainable parameters, suggesting that topology-level regularization of the mixing operator is a practical and efficient lever for long-context modeling. We view this spectral framing as complementary to positional extrapolation and kernel-efficiency advances, and hope it helps guide architectures that remain stable as sequence length increases.

## Acknowledgements

This work was supported in part by the New Generation Artificial Intelligence-National Science and Technology Major Project (No. 2025ZD0122702), the Shenzhen Medical Research Funds in China (No. B2302037), Natural Science Foundation of China (No. U24B6012, 62406167, 61972217, 32071459, 62176249, 62006133, 62271465), AI for Science (AI4S)-Preferred Program, Peking University Shenzhen Graduate School, China, and the Guangdong S&T Program (2024B0101010003).

## Impact Statement

This paper presents work whose goal is to advance the field of Machine Learning. There are many potential societal consequences of our work, none of which we feel must be specifically highlighted here.

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

# A. Detailed Experimental Setup

This appendix provides the complete implementation details, training configurations, and evaluation protocols used in our experiments.

## A.1. Datasets and Benchmarks

**Training Datasets.** We use three datasets, each tailored to a specific setting. *Alpagasus-5k* (Chen et al., 2024), a GPT-4-filtered high-quality subset of Alpaca, supports the general-capability evaluation (MMLU, HellaSwag) and all ablations, simulating a standard instruction-tuning scenario. *RULER* (Hsieh et al., 2024) supports post-training long-context extrapolation: we fine-tune on its training split with a fixed 4K context window so the model learns the task format within a standard window before being tested at 8K, 16K, and 32K. *NoLiMa* (Modarressi et al.) supports the from-scratch experiments. It minimizes lexical overlap between a question and its needle, so that answering requires latent associative reasoning rather than literal matching, relying on the world knowledge of a pretrained model to bridge the question and the needle. Since our 109M models are trained from scratch and lack such priors, we adapt the protocol into a self-contained corpus: for every instance (100% of both training and evaluation data) we explicitly insert the bridging fact into the haystack, turning each query into a multi-hop chain over distant inserted statements. This keeps the no-literal-matching property while drawing train and test instances from the same template pool, so that long-context failures reflect mechanistic spectral decay rather than distribution shift.

**Evaluation Benchmarks.** General capabilities are evaluated on MMLU (Hendrycks et al., 2021) and HellaSwag (Zellers et al., 2019) with models fine-tuned on Alpagasus-5k. For long-context extrapolation, RULER spans four task categories—Retrieval (NIAH), Multi-hop Tracing (Variable Tracking), Aggregation (Common/Frequent Words Extraction), and Question Answering, 13 tasks in total—and we evaluate at 8K, 16K, and 32K using models fine-tuned on RULER-4K. NoLiMa is evaluated up to 8K using the from-scratch models trained at 1K, reporting the exact-match accuracy of the retrieved associated value.

## A.2. Model Architectures

**Post-Training Setting (Llama-3.2).** We build on Llama-3.2-1B-Instruct (Dubey et al., 2024). The *Raw* baseline is the unmodified model. *LoRA* (Hu et al., 2022) adapts the query, key, value, and output projections ($W_q, W_k, W_v, W_o$) of the attention layers with rank $r = 8$ and scaling $\alpha = 16$. *LoRA + Gated Attention* adds, on top of LoRA, the gated-attention mechanism of Qiu et al. (2026): a query-dependent, head-specific sigmoid gate applied elementwise to the SDPA output, realized by a dense per-layer projection that adds $\approx 67M$ trainable parameters. *LoRA + TRSP (ours)* adds the triBox side-path and long-context gate in parallel to attention, $\mathbf{X}_{\text{out}} = \mathbf{X}_{\text{in}} + \text{Attn}(\mathbf{X}_{\text{in}}) + g_\ell \, \text{triBox}_{b_\ell}(\mathbf{X}_{\text{in}})$ (Eq. (6)); the only extra trainable parameters are the gate MLP and the scalar $\beta$ of Eq. (7) ($\approx 50$ in total).

**Pre-Training Setting (Custom Tiny Transformer).** To isolate architectural effects, we train a 109M-parameter Transformer from scratch whose configuration follows Llama at a smaller scale: hidden size $d_{\text{model}} = 512$, $D = 14$ layers, 8 attention heads, MLP ratio 4.0, a 32,000-token vocabulary (Llama tokenizer), and Rotary Positional Embeddings (Su et al., 2024). We compare four variants that share the same backbone and parameter count ($\approx 109.8M$): the standard Transformer, Transformer + Gated Attention (Qiu et al., 2026), the Differential Transformer (Ye et al., 2025), and Transformer + TRSP. Following the official Differential Transformer implementation, we partition the query and key projections into two groups to form two softmax maps whose difference is the attention score, $\text{Attn} = \text{softmax}(Q_1 K_1^\top) - \lambda \, \text{softmax}(Q_2 K_2^\top)$, where $\lambda$ is a per-layer learnable scalar shared across heads and reparameterized as $\lambda = \exp(\lambda_{q_1} \cdot \lambda_{k_1}) - \exp(\lambda_{q_2} \cdot \lambda_{k_2}) + \lambda_{\text{init}}$; we apply per-head GroupNorm to the head outputs and halve the number of heads so that the parameter count matches the standard Transformer.

## A.3. Training Configurations

All models use the AdamW optimizer with a cosine schedule and a 3% warmup.

**Post-training (Alpagasus & RULER).** For MMLU/HellaSwag and the ablations we fine-tune on Alpagasus-5k for 3 epochs with batch size 1 at a learning rate of $6 \times 10^{-4}$; the sensitivity study additionally sweeps $8 \times 10^{-4}$ and $1 \times 10^{-3}$. For RULER extrapolation we fine-tune on the RULER training set at a fixed 4K context for 3 epochs at $6 \times 10^{-4}$.

**Pre-training (NoLiMa).** We train the 109M models from random initialization on the augmented NoLiMa corpus with a

fixed 1K context for 5 epochs at $1 \times 10^{-4}$, using standard next-token prediction (causal language modeling).

### A.4. Evaluation Protocols

**Spectral metrics.** We compute the diagnostics of §4.1 on the layer-wise token representations $H_\ell \in \mathbb{R}^{T \times d}$ (reported in Table 1 and Figure 4). *Stable Rank* is $\|H_\ell\|_F^2 / \|H_\ell\|_2^2$, measuring the effective dimensionality. *Anisotropy* is the average cosine similarity between random pairs of token representations; lower values indicate a more isotropic distribution, which we treat as beneficial only when accompanied by high stable rank (structured isotropy). *Signal Propagation Rate (SPR)* is the perturbation gain $\|\delta_{\text{out}}\| / \|\delta_{\text{in}}\|$ (Gouk et al., 2021), measured by injecting a Gaussian perturbation ($\sigma = 10^{-3}$) at the embeddings and propagating it through the network. *Spectral Flatness* is the ratio of the geometric to the arithmetic mean of the singular values of $H_\ell$.

**Extrapolation testing.** RULER models trained at 4K are evaluated at 8K, 16K, and 32K, reporting the average score over all sub-tasks; NoLiMa models trained at 1K are evaluated at 1K, 2K, 4K, and 8K, reporting the exact-match accuracy of the generated answer against the gold reference.

## B. Detailed Breakdown of RULER Performance

We provide the fine-grained performance breakdown across all 13 sub-tasks of the RULER benchmark in Table 5. This detailed view reveals specific failure modes of baseline methods that are masked in the aggregated scores, particularly at the extreme context length of 32K.

**Analysis of Sub-Task Performance.** While the LoRA + Gated Attention baseline remains competitive at 8K, it exhibits a catastrophic collapse as the context extends to 32K. Specifically, on NIAH Multikey 2 and NIAH Multikey 3, its accuracy plummets to near zero (6.3% and 0.0% respectively). In contrast, our TRSP maintains high robustness, achieving 93.7% and 64.5% on these tasks, indicating that our spectral regularization helps prevent the attention-sink phenomenon from cutting off long-range dependencies.

Regarding the harder tasks like Common Words Extraction (CWE) and Variable Tracking (VT), which require precise state tracking over long distances, TRSP achieves 8.1% on CWE (vs. 0.0% for LoRA) and 18.1% on VT (vs. 9.2% for LoRA). This roughly doubles VT accuracy (18.1% vs. 9.2%), consistent with our account: by maintaining a healthier spectral gap and effective rank, TRSP better preserves the distinctness of token states over very long sequences. Standard LoRA degrades sharply at 32K, whereas TRSP mitigates this decay.

For full transparency, we also note the tasks where the raw instruction-tuned model retains a clear edge: on NIAH Multi-Value (NIAH MV), NIAH Multi-Query (NIAH MQ), and Frequent-Words Extraction, the raw model scores far higher (e.g., 80.6%, 84.7%, and 63.5% at 32K) than both LoRA and LoRA + TRSP, which regress to roughly 25–33%. Because this regression is shared almost identically by LoRA and LoRA + TRSP, it reflects a format/distribution shift induced by RULER fine-tuning rather than a side-effect of the TRSP branch; TRSP's gains instead concentrate on the retrieval- and tracking-heavy tasks (NIAH MK2/MK3, CWE, VT) that most directly stress long-range spectral health.

## C. Theoretical Analysis of TRSP Topology

This appendix analyzes the spectral properties of the TRSP mixing operator, establishing (i) a length-independent lower bound on its algebraic connectivity and (ii) a lower bound on its effective rank. Throughout, $\Omega(\cdot)$ and $\Theta(\cdot)$ denote asymptotics in the sequence length $T$, all constants are independent of $T$ unless stated otherwise, and modeling assumptions are made explicit where they are used.

### C.1. Setup and Definitions

We study the per-channel token-mixing operator induced by one TRSP layer. Acting on the token axis, a single channel of the hidden state is $x \in \mathbb{R}^T$, and the (linear) layer update is

$$x_{\text{out}} = \left(I + A_{\text{attn}} + g\, M_{\text{tri}}\right) x =: \mathcal{M}\, x, \tag{8}$$

where $A_{\text{attn}} \in \mathbb{R}^{T \times T}$ is the row-stochastic attention matrix, $M_{\text{tri}} \in \mathbb{R}^{T \times T}$ is the triBox operator, and $g := g_\ell \in (0, 1)$ is the long-context gate of Eq. (7) (fixed within a forward pass).

*Table 5.* **Detailed RULER Sub-Task Performance (8K, 16K, 32K).** We report accuracy (%) for each task. Raw: Llama-3.2-1B-Instruct. Gate: LoRA + Gated Attention. Ours: LoRA + TRSP. Best results in each group are bolded.

| Sub-Task | Context Length: 8K | | | | Context Length: 16K | | | | Context Length: 32K | | | |
|---|---|---|---|---|---|---|---|---|---|---|---|---|
| | Raw | LoRA | Gated | **Ours** | Raw | LoRA | Gated | **Ours** | Raw | LoRA | Gated | **Ours** |
| NIAH Single 1 | 87.3 | **100.0** | 81.0 | **100.0** | 65.1 | **100.0** | 61.9 | **100.0** | 92.1 | **100.0** | 98.4 | **100.0** |
| NIAH Single 2 | **100.0** | **100.0** | **100.0** | **100.0** | 98.4 | **100.0** | 98.4 | **100.0** | 93.5 | 98.4 | 98.4 | **100.0** |
| NIAH Single 3 | 98.4 | **100.0** | 79.4 | 98.4 | 96.8 | **100.0** | 68.2 | **100.0** | **100.0** | **100.0** | **100.0** | **100.0** |
| NIAH MK 1 | 95.2 | 96.8 | **100.0** | **100.0** | 91.9 | 96.8 | 95.2 | **98.4** | 83.9 | 93.5 | **100.0** | **100.0** |
| NIAH MK 2 | 88.7 | 91.9 | 66.1 | **100.0** | 75.8 | 85.5 | 56.5 | **100.0** | 90.5 | 74.6 | 6.3 | **93.7** |
| NIAH MK 3 | 12.9 | 96.8 | 85.5 | **98.4** | 14.5 | 85.5 | 59.7 | **88.7** | 27.4 | 62.9 | 0.0 | **64.5** |
| NIAH MV | **86.5** | 25.0 | 23.8 | 25.0 | **82.1** | 25.0 | 23.0 | 24.6 | **80.6** | 24.6 | 24.6 | 25.0 |
| NIAH MQ | **93.7** | 25.0 | 20.2 | 25.0 | **94.4** | 25.0 | 20.2 | 25.0 | **84.7** | 23.8 | 23.4 | 25.0 |
| Variable Track | **32.3** | 20.0 | 19.4 | 20.0 | 3.2 | 12.6 | 19.0 | **19.7** | 4.4 | 9.2 | 1.9 | **18.1** |
| Common Words | 0.2 | 0.2 | 0.8 | **6.6** | 0.5 | 0.0 | 0.0 | **8.7** | 0.8 | 0.0 | 0.2 | **8.1** |
| Freq Words | **63.0** | 33.3 | 1.6 | 33.3 | **78.3** | 32.8 | 9.5 | 32.8 | **63.5** | 32.8 | 30.7 | 32.3 |
| QA 1 | 40.3 | 59.7 | 6.5 | **71.0** | 37.1 | **51.6** | 3.2 | 45.2 | 30.6 | 51.6 | 12.9 | **53.2** |
| QA 2 | 31.8 | **81.0** | 1.6 | 76.2 | 31.8 | 68.2 | 0.0 | **71.4** | 22.2 | **76.2** | 27.0 | 65.1 |

**Symmetrization.** The causal triBox $M_{\text{tri}}$ is lower-triangular (directed). Algebraic connectivity is defined for undirected graphs, so we analyze the symmetrized connection $W_{\text{tri}} := \frac{1}{2}(M_{\text{tri}} + M_{\text{tri}}^\top)$ (and likewise $W_{\text{attn}}$); both are symmetric with nonnegative weights. We write $\mathbf{L}_\bullet = \mathbf{D}_\bullet - W_\bullet$ for the corresponding combinatorial Laplacian, with $\mathbf{D}_\bullet$ the diagonal degree matrix; each such Laplacian is symmetric positive semidefinite (PSD) with smallest eigenvalue 0.

**Two spectral quantities.** We distinguish (i) the *algebraic connectivity* $\mu_2(\mathbf{L})$, the second-smallest eigenvalue of $\mathbf{L}$ (the Fiedler value), measuring absolute connectivity; and (ii) the *normalized gap* $\gamma$, the second-smallest eigenvalue of the random-walk Laplacian $\mathcal{L} = I - \mathbf{D}^{-1}W$, measuring the per-step contraction rate of the induced random walk. For an $r$-regular graph the two satisfy $\gamma = \mu_2/r$.

## C.2. Topological Structure: A Cyclic Cayley Graph

We first identify the connectivity that the triBox branch induces across layers.

**Theorem C.1** (triBox skeleton as a Cayley graph). *Consider the symmetrized connection obtained by superposing the dominant dyadic offsets of the triBox branch across layers $\ell = 0, \ldots, L$ with $L = \lfloor \log_2 T \rfloor$. Its connection graph is the Cayley graph $\text{Cay}(\mathbb{Z}_T, S)$ on the cyclic group $\mathbb{Z}_T$ with generator set $S = \{\pm 2^\ell : \ell = 0, \ldots, L\}$.*

*Proof.* At layer $\ell$ the triBox window of bandwidth $b_\ell = 2^\ell$ couples each token to its causal neighborhood, whose longest offset is $2^\ell$. Retaining these dominant offsets and symmetrizing, the superposition over layers connects every $n \in \mathbb{Z}_T$ to $n \pm 2^\ell$ for all $\ell$, i.e. exactly $S$. The connection depends only on the difference $(n - n') \mod T$, so the symmetrized adjacency $W_{\text{tri}}$ is circulant—the defining property of a Cayley graph on $\mathbb{Z}_T$ (Biggs, 1993). □

*Remark* C.2 (Sparse skeleton versus full band). The full triBox couples each token to a contiguous band rather than to the single offset $2^\ell$; the band only adds shorter-range edges on top of $S$. Since adding edges can only increase the Laplacian eigenvalues (Theorem C.4), every lower bound on the algebraic connectivity proved for the sparse skeleton $\text{Cay}(\mathbb{Z}_T, S)$ also holds for the actual (denser) banded connectivity. We therefore analyze the skeleton, without loss of generality for lower bounds.

**Lemma C.3** (Spectrum of the skeleton). *For $\text{Cay}(\mathbb{Z}_T, S)$ with $S = \{\pm 2^\ell\}_{\ell=0}^L$, the combinatorial Laplacian $\mathbf{L}_{\text{tri}}$ is circulant with eigenvalues, indexed by Fourier modes $k \in \{0, \ldots, T-1\}$,*

$$\mu_k(\mathbf{L}_{\text{tri}}) = 2\sum_{\ell=0}^{L}\left(1 - \cos\left(2^\ell \theta_k\right)\right), \qquad \theta_k = \frac{2\pi k}{T}. \tag{9}$$

*Proof.* The eigenvectors of a symmetric circulant matrix are the Fourier modes $v_k = (e^{\mathrm{i}\,2\pi kn/T})_{n=0}^{T-1}$ (Biggs, 1993). Each

generator pair $\{+2^\ell, -2^\ell\}$ contributes degree 2 and off-diagonal phases $e^{\pm i2^\ell \theta_k}$, hence a Laplacian eigenvalue contribution $2 - \left(e^{i2^\ell \theta_k} + e^{-i2^\ell \theta_k}\right) = 2\left(1 - \cos(2^\ell \theta_k)\right)$. Summing over $\ell$ gives the claim. $\qquad\square$

## C.3. Bounded Spectral Gap

### C.3.1. SAFETY NET: MONOTONICITY UNDER PARALLEL COMPOSITION

**Lemma C.4** (Weyl monotonicity). *Let $P, Q$ be symmetric with $Q \succeq 0$. Then $\lambda_k(P + Q) \geq \lambda_k(P)$ for every $k$, where eigenvalues are listed in increasing order.*

*Proof.* By the Courant–Fischer min–max theorem, $\lambda_k(P + Q) = \min_{\dim V = k} \max_{0 \neq v \in V} \frac{v^\top (P+Q)v}{v^\top v} \geq \min_{\dim V = k} \max_{0 \neq v \in V} \frac{v^\top P v}{v^\top v} = \lambda_k(P)$, where the inequality uses $v^\top Q v \geq 0$. $\qquad\square$

**Proposition C.5** (Safety net). *The symmetrized composite connection has Laplacian $\mathbf{L} = \mathbf{L}_{\text{attn}} + g\,\mathbf{L}_{\text{tri}}$, and its algebraic connectivity satisfies $\mu_2(\mathbf{L}) \geq g\,\mu_2(\mathbf{L}_{\text{tri}})$ for any attention pattern.*

*Proof.* The Laplacian of a weighted union of edge sets is the sum of the individual Laplacians, so $\mathbf{L} = \mathbf{L}_{\text{attn}} + g\,\mathbf{L}_{\text{tri}}$ with $\mathbf{L}_{\text{attn}} \succeq 0$. Applying Theorem C.4 with $P = g\,\mathbf{L}_{\text{tri}}$, $Q = \mathbf{L}_{\text{attn}}$ at $k = 2$ gives $\mu_2(\mathbf{L}) = \lambda_2(\mathbf{L}) \geq \lambda_2(g\,\mathbf{L}_{\text{tri}}) = g\,\mu_2(\mathbf{L}_{\text{tri}})$. $\qquad\square$

Thus, even if attention disconnects ($\mu_2(\mathbf{L}_{\text{attn}}) \to 0$), the composite connectivity stays $\geq g\,\mu_2(\mathbf{L}_{\text{tri}}) > 0$: the side-path is a "safety net".

### C.3.2. CONSTANT ALGEBRAIC CONNECTIVITY OF THE TRIBOX SKELETON

**Theorem C.6** (Length-independent connectivity). *For every $T \geq 2$, the triBox skeleton satisfies $\mu_2(\mathbf{L}_{\text{tri}}) \geq 2$; in particular it is bounded below by a positive constant independent of $T$.*

*Proof.* By Theorem C.3 it suffices to show that for every nonzero mode $k \in \{1, \ldots, T - 1\}$ there is some $\ell \in \{0, \ldots, L\}$ with $1 - \cos(2^\ell \theta_k) \geq 1$, i.e. $2^\ell \theta_k \bmod 2\pi \in [\frac{\pi}{2}, \frac{3\pi}{2}]$; that single term then gives $\mu_k \geq 2$.

Write $\psi := k/T \in (0, 1)$ with binary expansion $\psi = \sum_{j \geq 1} b_j 2^{-j}$, $b_j \in \{0, 1\}$. Then $2^\ell \theta_k \bmod 2\pi = 2\pi \{2^\ell \psi\}$ and $\{2^\ell \psi\} = \sum_{j \geq 1} b_{\ell+j} 2^{-j}$ has leading bits $(b_{\ell+1}, b_{\ell+2})$. A direct check shows $\{2^\ell \psi\} \in [\frac{1}{4}, \frac{3}{4})$ iff $b_{\ell+1} \neq b_{\ell+2}$, which gives $2\pi \{2^\ell \psi\} \in [\frac{\pi}{2}, \frac{3\pi}{2})$ and hence $1 - \cos \geq 1$.

It remains to find a sign change $b_{\ell+1} \neq b_{\ell+2}$ with $\ell \in \{0, \ldots, L\}$, i.e. among the first $L + 2$ bits of $\psi$. If there were none, then $b_1 = \cdots = b_{L+2}$: all 0 forces $\psi < 2^{-(L+2)}$, all 1 forces $\psi \geq 1 - 2^{-(L+2)}$. But $1 \leq k \leq T - 1$ gives $\psi \in [\frac{1}{T}, 1 - \frac{1}{T}]$, and $L = \lfloor \log_2 T \rfloor$ yields $2^{-(L+2)} = 2^{-L}/4 < 1/T$ (since $2^{-L} < 2/T$), contradicting both cases. Hence a sign change exists and $\mu_k \geq 2$ for all $k \neq 0$, so $\mu_2(\mathbf{L}_{\text{tri}}) = \min_{k \neq 0} \mu_k \geq 2$. $\qquad\square$

### C.3.3. NORMALIZED GAP AND THE SPARSE/DENSE TRADE-OFF

We first note $\mu_2(\mathbf{L}_{\text{tri}}) = \Theta(1)$: the lower bound $\geq 2$ is Theorem C.6, and for the matching upper bound the mode $k = 1$ (with $\theta_1 = 2\pi/T$) gives, via $1 - \cos x \leq x^2/2$,

$$\mu_2 \leq \mu_1 = 2 \sum_{\ell=0}^{L} \left(1 - \cos(2^\ell \theta_1)\right) \leq \theta_1^2 \sum_{\ell=0}^{L} 4^\ell \leq \left(\frac{2\pi}{T}\right)^2 \frac{4^{L+1}}{3} \leq \frac{16\pi^2}{3}, \tag{10}$$

using $4^{L+1} \leq 4T^2$ from $2^L \leq T$. The skeleton is $r$-regular with degree $r = |S| = 2(L + 1) = \Theta(\log T)$, so by $\gamma = \mu_2/r$,

$$\gamma_{\text{tri}} = \frac{\mu_2(\mathbf{L}_{\text{tri}})}{r} = \Theta\left(\frac{1}{\log T}\right). \tag{11}$$

The *unnormalized* connectivity is thus constant while the *normalized* (per-step) gap decays as $\Theta(1/\log T)$. Two implementations trade these off. *Dense (ablation).* Each layer realizes all offsets $2^0, \ldots, 2^{D-1}$, so a single layer attains $\gamma = \Theta(1/\log T)$; if additionally the depth scales as $D = \Theta(\log T)$, the depth-composed contraction is

$(1 - \gamma)^D = (1 - \Theta(1/\log T))^{\Theta(\log T)} = \Theta(1)$ (a constant global normalized gap), at the cost of $O(Td \log T)$ per-layer work. *Standard (TRSP)*. The offsets are distributed across layers, so the Cayley skeleton forms only globally by superposition; this retains the constant unnormalized connectivity $\mu_2 = \Omega(1)$ (Theorem C.6) at $O(Td)$ cost, while the global normalized gap stays $\Theta(1/\log T)$. We adopt the standard implementation: by Theorem C.5 the constant unnormalized connectivity already rules out the disconnection (isolation) failure mode, and the dense variant serves only as a theoretical upper bound. The assumption $D = \Theta(\log T)$ is used only for the dense variant.

### C.4. Effective Rank Lower Bound

We use the stable rank as the effective-rank surrogate (consistent with §3.2), $R_{\text{eff}}(\mathcal{M}) = \|\mathcal{M}\|_F^2 / \|\mathcal{M}\|_2^2$.

**Proposition C.7** (Frobenius energy). *Let the triBox at the layer of interest have bandwidth b. Then*

$$\|\mathcal{M}\|_F^2 \geq T + g^2 \|M_{\text{tri}}\|_F^2 = T(1 + g^2 E_b), \tag{12}$$

*where $E_b := \|M_{\text{tri}}\|_F^2 / T$ is the per-row energy of the normalized triangular kernel* **h**, *satisfying $E_b = \|\mathbf{h}\|_2^2 = \frac{2b^2 + 1}{3b^3} = \Theta(1/b)$. In particular $E_b = \Theta(1)$ for proximal layers with bounded bandwidth $b = O(1)$.*

*Proof.* Write $\mathcal{M} = I + B$ with $B = A_{\text{attn}} + g M_{\text{tri}}$. Then $\|\mathcal{M}\|_F^2 = \|I\|_F^2 + 2\langle I, B \rangle + \|B\|_F^2 = T + 2\operatorname{tr}(B) + \|B\|_F^2$. Both $A_{\text{attn}}$ and $M_{\text{tri}}$ are entrywise nonnegative, so $\operatorname{tr}(B) \geq 0$; expanding $\|B\|_F^2 = \|A_{\text{attn}}\|_F^2 + 2g\langle A_{\text{attn}}, M_{\text{tri}} \rangle + g^2 \|M_{\text{tri}}\|_F^2$ with the entrywise cross term $\langle A_{\text{attn}}, M_{\text{tri}} \rangle \geq 0$ gives $\|B\|_F^2 \geq g^2 \|M_{\text{tri}}\|_F^2$. Hence $\|\mathcal{M}\|_F^2 \geq T + g^2 \|M_{\text{tri}}\|_F^2$. For the kernel energy, each (interior) row of $M_{\text{tri}}$ is the normalized triangular kernel **h** with $h(r) = (b - |r - (b-1)|)/b^2$ for $r = 0, \ldots, 2b - 2$; a direct computation gives

$$\|\mathbf{h}\|_2^2 = \frac{1}{b^4} \sum_{m=-(b-1)}^{b-1} (b - |m|)^2 = \frac{1}{b^4} \cdot \frac{b(2b^2 + 1)}{3} = \frac{2b^2 + 1}{3b^3} = \Theta(1/b). \tag{13}$$

Summing the $T$ rows, $\|M_{\text{tri}}\|_F^2 = T \cdot \Theta(1/b)$, so $E_b = \Theta(1/b)$. $\square$

**Theorem C.8** (Effective rank does not collapse at proximal layers). *Consider a proximal layer with bounded bandwidth $b = O(1)$ in $T$, and suppose attention degenerates to a rank-one sink with $\|A_{\text{attn}}\|_2 = \Theta(\sqrt{T})$ (e.g. $A_{\text{attn}} = \mathbf{1}e_1^\top$). Then*

$$\liminf_{T \to \infty} R_{\text{eff}}(\mathcal{M}) \geq 1 + g^2 E_b > 1. \tag{14}$$

*Proof.* The banded row-stochastic $M_{\text{tri}}$ has $\|M_{\text{tri}}\|_\infty = 1$ (row sums) and $\|M_{\text{tri}}\|_1 = O(\log b)$ (column sums, the harmonic factor arising only at the first $O(b)$ boundary columns), so $\|M_{\text{tri}}\|_2 \leq \sqrt{\|M_{\text{tri}}\|_1 \|M_{\text{tri}}\|_\infty} = O(\sqrt{\log b}) = o(\sqrt{T})$. By the triangle inequality $\|\mathcal{M}\|_2 \leq \|I\|_2 + \|A_{\text{attn}}\|_2 + g\|M_{\text{tri}}\|_2 = \sqrt{T} + o(\sqrt{T})$, so $\|\mathcal{M}\|_2^2 \leq T + o(T)$. Combining with Theorem C.7,

$$R_{\text{eff}}(\mathcal{M}) = \frac{\|\mathcal{M}\|_F^2}{\|\mathcal{M}\|_2^2} \geq \frac{T(1 + g^2 E_b)}{T + o(T)} = \frac{1 + g^2 E_b}{1 + o(1)} \xrightarrow[T \to \infty]{} 1 + g^2 E_b. \tag{15}$$

Since $g > 0$ and $E_b = \Theta(1) > 0$ for bounded $b$, the limit exceeds 1: at proximal layers the side-path keeps the effective rank bounded away from the rank-one value to which a pure sink collapses. For wide (distal) layers $b = \Theta(T)$ one has $E_b = \Theta(1/T)$, so the side-path there contributes to global mixing (Section C.3) rather than to rank preservation—precisely the proximal/distal division of §3.2. $\square$

*Remark* C.9 (Ideal normalization). The numerator bound $\|\mathcal{M}\|_F^2 \geq T$ holds for any layer (already from the identity term). Hence if output normalization bounds $\|\mathcal{M}\|_2 \leq C_{\max} = \Theta(1)$, then $R_{\text{eff}}(\mathcal{M}) \geq \|\mathcal{M}\|_F^2 / C_{\max}^2 = \Omega(T)$, so the operator can use a dimensionality that grows with the sequence length.

## D. Theoretical Analysis of Spectral Properties

This appendix derives a *sufficiency* result: under explicit spectral conditions, the inference error of a class of global reasoning tasks admits an upper bound that decreases as the spectral gap and the effective rank increase. The argument is a worst-case bound rather than an exact characterization, and we state every assumption where it is used.

**Operator under analysis.** Consistent with §1, the object here is the *row-normalized transition operator $M$* obtained by row-stochastic normalization of the residual operator $\mathcal{M}$ of Section C; it admits a stationary distribution $\pi$ and acts as a Markov mixing operator, the standard setting for contraction analysis (Levin & Peres, 2017). For the spectral-gap step we assume $M$ is *reversible* (equivalently, we analyze its $\pi$-reversibilization $\frac{1}{2}(M + M^{*\pi})$, with $M^{*\pi}$ the adjoint in $\langle \cdot, \cdot \rangle_\pi$), so that $M$ is self-adjoint in $\langle \cdot, \cdot \rangle_\pi$ with real spectrum in $[-1, 1]$ and leading eigenvector $\mathbf{1}$. Since $M$ and $\mathcal{M}$ share the same connectivity, the gap and rank guarantees of Section C carry over; we work in the $\pi$-weighted norm $\|u\|_{2,\pi}^2 := \sum_i \pi_i u_i^2$.

### D.1. Assumptions and Setup

Let $d$ be the feature dimension and $L$ the network depth. We write $X_\ell \in \mathbb{R}^{T \times d}$ for the state at layer $\ell$. We model the prediction as $\hat{y} = \mathcal{N}(X) = G(M^L X^{(0)})$, where $X^{(0)}$ is the injected input, $M^L$ is the depth-$L$ mixing, and $G(z) = f(\mathcal{A}^{-1} z)$ is the readout with linear part $\mathcal{A} : \mathbb{R}^d \to \mathbb{R}^d$; the target is $y = f(X)$ with $L_f := \text{Lip}(f)$, and the inference error is $\mathcal{E} := \|\hat{y} - y\|$.

- **(A1) Reversible spectral gap.** $M$ is reversible with stationary $\pi$ ($M\mathbf{1} = \mathbf{1}$, self-adjoint in $\langle \cdot, \cdot \rangle_\pi$) and contracts on the zero-mean subspace $\mathcal{H}_0(\pi) = \{v : \sum_i \pi_i v_i = 0\}$: $\|Mv\|_{2,\pi} \leq (1 - \gamma)\|v\|_{2,\pi}$ for all $v \in \mathcal{H}_0(\pi)$, with $\gamma \in (0, 1)$.

- **(A2) Non-degenerate stationary distribution.** $\pi_{\min} \geq c_\pi / T$ for a constant $c_\pi > 0$.

- **(A3) High effective rank.** The stable rank satisfies $\text{sr}(X_\ell) = \|X_\ell\|_F^2 / \|X_\ell\|_2^2 \geq r_{\min}$.

- **(A4) Well-conditioned readout.** $\mathcal{A}$ is invertible with $\kappa(\mathcal{A}) \leq \bar{\kappa}$, and the residual structure keeps $\sigma_{\max}(\mathcal{A}) \geq c_0 > 0$.

- **(A5) Distributed, isotropic readout.** The target is recovered by aggregating evidence across the $k = \Theta(\text{sr}(X_\ell))$ well-conditioned feature directions of Theorem D.1, with task-irrelevant components that are uncorrelated across these directions and have per-direction variance at most $\sigma_\perp^2$.

The proof combines four lemmas: Theorem D.1 (feature injection, from A3/A5), Theorem D.3 (exponential mixing, from A1), Theorem D.4 (pointwise alignment, from A2), and Theorem D.5 (readout stability, from A4), assembled in Section D.6.

### D.2. Lemma 1: Non-degenerate, Rank-rich Feature Injection

**Lemma D.1** (Restricted invertibility of the state). *Under (A3), for any $\varepsilon \in (0, 1)$ there is a feature subspace $S$ (a subset of the $d$ feature coordinates) of dimension $k \geq (1 - \varepsilon)^2 \text{sr}(X_\ell) \geq (1 - \varepsilon)^2 r_{\min}$ on which the state map is uniformly well-conditioned:*

$$\sigma_{\min}(X_\ell|_S) \geq \varepsilon \frac{\|X_\ell\|_2}{\sqrt{d}} =: \lambda_{\text{low}} > 0. \tag{16}$$

*Consequently the restricted map is left-invertible with $\|(X_\ell|_S)^\dagger\|_2 \leq 1/\lambda_{\text{low}}$.*

*Proof.* This is the Spielman–Srivastava form of the Bourgain–Tzafriri restricted invertibility theorem (Bourgain & Tzafriri, 1987; Spielman & Srivastava, 2012), applied to $X_\ell \in \mathbb{R}^{T \times d}$ with its $d$ columns as feature directions: for any $\varepsilon \in (0, 1)$ there is a column subset $S$ with $|S| \geq (1 - \varepsilon)^2 \|X_\ell\|_F^2 / \|X_\ell\|_2^2 = (1 - \varepsilon)^2 \text{sr}(X_\ell)$ and $\sigma_{\min}(X_\ell|_S) \geq \varepsilon \|X_\ell\|_2 / \sqrt{d}$. Assumption (A3) lower-bounds the size as $|S| \geq (1 - \varepsilon)^2 r_{\min}$, and the singular-value bound gives the stated pseudo-inverse norm. $\square$

The lemma plays two roles. Its singular-value floor $\lambda_{\text{low}} > 0$ certifies that the feature injection is *non-degenerate* (the readout can invert it stably); its dimension count shows that the number of well-conditioned, independently usable directions grows linearly with the effective rank. The latter is what the effective rank buys, and it drives the following error reduction.

**Corollary D.2** (Rank-driven readout averaging). *Under (A3) and (A5), the readout error contributed by the task-irrelevant component is at most $\sigma_\perp / \sqrt{k} = O(\sigma_\perp / \sqrt{\text{sr}(X_\ell)})$.*

*Proof.* By Theorem D.1 the target is recovered from $k \geq (1 - \varepsilon)^2 \text{sr}(X_\ell)$ well-conditioned directions. Aggregating $k$ uncorrelated, zero-mean components of per-direction variance at most $\sigma_\perp^2$ (A5) yields an estimator whose irrelevant-component variance is at most $\sigma_\perp^2 / k$; taking square roots gives error $\leq \sigma_\perp / \sqrt{k} = O(\sigma_\perp / \sqrt{\text{sr}(X_\ell)})$. $\square$

### D.3. Lemma 2: Exponential Mixing

**Lemma D.3.** *Under (A1), the depth-$L$ operator satisfies $\|M^L\|_{\mathcal{H}_0 \to \mathcal{H}_0} \leq (1-\gamma)^L$, where $\|\cdot\|_{\mathcal{H}_0 \to \mathcal{H}_0}$ is the operator norm induced by $\|\cdot\|_{2,\pi}$ on the zero-mean subspace.*

*Proof.* By (A1), $M$ is self-adjoint in $\langle\cdot,\cdot\rangle_\pi$ with $M\mathbf{1} = \mathbf{1}$, so its orthogonal complement $\mathcal{H}_0(\pi) = \mathbf{1}^{\perp_\pi}$ is $M$-invariant; (A1) then gives $\|M\|_{\mathcal{H}_0 \to \mathcal{H}_0} = \sup_{0 \neq v \in \mathcal{H}_0} \|Mv\|_{2,\pi}/\|v\|_{2,\pi} \leq 1 - \gamma$. By submultiplicativity of the operator norm over the $L$ layers,

$$\|M^L\|_{\mathcal{H}_0 \to \mathcal{H}_0} \leq \prod_{\ell=1}^{L} \|M\|_{\mathcal{H}_0 \to \mathcal{H}_0} \leq (1-\gamma)^L. \tag{17}$$

$\square$

### D.4. Lemma 3: Pointwise Error Alignment

**Lemma D.4.** *Under (A2), any error vector $e$ satisfies $\|e\|_\infty \leq \sqrt{T/c_\pi}\,\|e\|_{2,\pi}$.*

*Proof.* Let $i^\star = \arg\max_i |e_i|$, so $|e_{i^\star}| = \|e\|_\infty$. Then

$$\|e\|_{2,\pi}^2 = \sum_{i=1}^{T} \pi_i e_i^2 \geq \pi_{i^\star} e_{i^\star}^2 \geq \pi_{\min} \|e\|_\infty^2 \geq \frac{c_\pi}{T} \|e\|_\infty^2, \tag{18}$$

using (A2) in the last step. Rearranging gives $\|e\|_\infty \leq \sqrt{T/c_\pi}\,\|e\|_{2,\pi}$. $\square$

### D.5. Lemma 4: Readout Stability

**Lemma D.5.** *Under (A4), the readout $G(z) = f(\mathcal{A}^{-1}z)$ satisfies $\mathrm{Lip}(G) \leq L_f\,\bar\kappa/c_0$.*

*Proof.* By the chain rule, $\mathrm{Lip}(G) \leq \mathrm{Lip}(f)\|\mathcal{A}^{-1}\|_2 = L_f/\sigma_{\min}(\mathcal{A}) = L_f\,\kappa(\mathcal{A})/\sigma_{\max}(\mathcal{A})$. Using $\kappa(\mathcal{A}) \leq \bar\kappa$ and $\sigma_{\max}(\mathcal{A}) \geq c_0$ from (A4) gives $\mathrm{Lip}(G) \leq L_f\bar\kappa/c_0$. The lower bound $\sigma_{\max}(\mathcal{A}) \geq c_0 > 0$ reflects that the residual (identity-skip) structure keeps the mean squared activation length from vanishing, so the forward map does not contract to zero (Hanin & Rolnick, 2018). $\square$

### D.6. Sufficiency: the Error Bound

We assemble the lemmas into a bound on $\mathcal{E} = \|\hat{y} - y\|$. The argument is a worst-case sketch: we bound the error by the product of how sensitively the readout reacts to its input and how far the mixed state is from the fully mixed (stationary) component,

$$\mathcal{E} \leq \underbrace{\left[\mathrm{Lip}(G) \cdot \rho_{\mathrm{avg}}\right]}_{\text{readout sensitivity}} \cdot \underbrace{\left\|M^L X^{(0)} - \bar{X}\right\|}_{\text{mixing residual}}, \tag{19}$$

where $\bar{X}$ is the $\pi$-stationary (fully mixed) component and $\rho_{\mathrm{avg}} = O(1/\sqrt{\mathrm{sr}(X_\ell)})$ is the readout-averaging gain of Theorem D.2.

*Readout sensitivity.* By Theorem D.5, $\mathrm{Lip}(G) \leq L_f\bar\kappa/c_0$; by Theorem D.2, aggregating over the $k = \Theta(\mathrm{sr}(X_\ell))$ well-conditioned directions of Theorem D.1 contributes $\rho_{\mathrm{avg}} = O(1/\sqrt{\mathrm{sr}(X_\ell)})$. Hence the readout sensitivity is $O\big(\kappa(\mathcal{A})/\sqrt{\mathrm{sr}(X_\ell)}\big)$.

*Mixing residual.* By Theorem D.3, $\|M^L X^{(0)} - \bar{X}\|_{2,\pi} \leq (1-\gamma)^L \|X^{(0)} - \bar{X}\|_{2,\pi}$; by Theorem D.4 and (A2), the pointwise residual obeys $\|M^L X^{(0)} - \bar{X}\|_\infty \leq \sqrt{T/c_\pi}\,(1-\gamma)^L \|X^{(0)} - \bar{X}\|_{2,\pi}$.

Combining the two factors, for constants $C_1, C_2 > 0$ (absorbing $L_f, \bar\kappa, c_0, \sigma_\perp$ and $\|X^{(0)} - \bar{X}\|_{2,\pi}$),

$$\boxed{\mathcal{E} \leq \underbrace{\frac{C_1\,\kappa(\mathcal{A})}{\sqrt{\mathrm{sr}(X_\ell)}}}_{\text{stability (A3–A5)}} \cdot \underbrace{C_2\sqrt{\frac{T}{c_\pi}}\,(1-\gamma)^L}_{\text{mixing (A1–A2)}}.} \tag{20}$$

The bound decreases as the effective rank $\mathrm{sr}(X_\ell)$ and the spectral gap $\gamma$ increase and as the condition number $\kappa(\mathcal{A})$ decreases—precisely the spectral quantities that TRSP regularizes. The factor $\sqrt{T/c_\pi}$ is the worst-case amplification from the $\pi$-weighted $L_2$ norm to the pointwise norm (Theorem D.4); when (A2) is tight, $\pi_{\min} = \Theta(1/T)$ and this factor is of order $1/\sqrt{\pi_{\min}}$. Consequently the bound is small only when the depth-driven contraction $(1 - \gamma)^L$ dominates the $\sqrt{T}$ amplification, i.e. when the gap is preserved as length grows. We emphasize that this is a worst-case sufficiency bound under (A1)–(A5); in particular the $1/\sqrt{\mathrm{sr}(X_\ell)}$ improvement relies on the distributed-readout assumption (A5), whereas restricted invertibility (Theorem D.1) alone guarantees only the non-degeneracy floor $\lambda_{\mathrm{low}} > 0$.

