# OpenReview forum: "The Devil is in the Spectrum: Mitigating Representation Collapse in LLMs via Topologically Regularized Side-Path"
_ICML.cc/2026/Conference — ICML 2026 regular_

### Official Review · Reviewer_8x8k · 2026-03-13

**Soundness:** 3
**Presentation:** 4
**Significance:** 3
**Originality:** 2
**Overall Recommendation:** 4
**Confidence:** 3

**Summary:**

The paper looks at representational collapse in large language models (LLMs). They first present a distinction between homogenization and isolation collapse. They attribute the collapse to models' inability to balance information mixing with information content (as measured by the effective rank). Their solution, TRSP, regularizes the token interaction topology to avoid the irregularities in attention patterns they identify as the cause of collapse: overmixing and attention sinks for homogenization collapse, and local attention for isolation collapse.

**Compliance With Llm Reviewing Policy:**

Affirmed.

**Final Justification:**

The authors added results that show a better link between their theoretical contribution and empirical performance

**Key Questions For Authors:**

The paper is clear and the problem is well presented. My main question is: how exactly does representation collapse cause issues, and what are these issues? while the story for how representation collapse happens, what forms it takes and how your solution solves them is clear, there is a missing link between that and performance issues. It would be great to have more details on how homogenization collapse degrades performance as context length increases for instance. This is particularly important since this form of representation collapse is usually associated with depth instead of context length.

**Limitations:**

yes

**Strengths And Weaknesses:**

Strengths
1. A clear description of the various mechanisms that lead to the collapse forms they identified. Figure 1 is particularly nice for getting the broader picture
2. The guarantee of high rank attention is useful to have

Weaknesses
1. It is not clear that representation collapse is a major limitation, and it would have been nice to see a bit more effort put into showing why it is
2. Results on models trained from scratch. The NoLiMa results are very useful, but not enough.
3. For the LoRA experiments, the solution does not seem to meaningfully reduce the performance drop associated with increased context lengths
4. While the authors mention the connection to oversmoothing in graph neural networks, they barely cite work that has been done on oversmoothing in transformers

---

> ### Author Rebuttal · Authors · 2026-03-31
>
> We sincerely thank the reviewer for the constructive feedback and for acknowledging the clarity of our presentation. We address each concern below.
>
> > Q1: How exactly does representation collapse cause performance issues?
>
> **Representation collapse directly destroys two computational prerequisites for long-context tasks: token distinguishability and long-range information flow.**
>
> **Homogenization collapse breaks token discrimination.** When effective rank drops, token representations converge to a narrow cone. This makes it impossible for the model to distinguish a target token from its distractors. Our RULER breakdown (in Table 6, Appendix B) provides direct evidence: at 32K, LoRA's accuracy on NIAH Multikey-2 (which requires distinguishing multiple needles) drops to 74.6% while TRSP maintains 93.7%, and on Variable Tracking (which requires maintaining distinct token states), LoRA drops to 9.2% vs. TRSP's 18.1%.
>
> **Isolation collapse severs long-range information flow.** When the spectral gap vanishes, distant tokens cannot exchange information. On NoLiMa (Table 3), which requires latent associative reasoning across the full context, the standard Transformer drops from 100% (1K) to 23.8% (8K), while TRSP retains 83.2% by maintaining a healthy spectral gap.
>
> **Empirical causal evidence.** Figures 4–5 show that as context grows from 1K to 8K, four independent spectral metrics degrade monotonically across baselines and accuracy drops in lockstep—while TRSP alone preserves both.
>
> **On depth vs. context length.** Recent work [1, 2] extends classical depth-focused rank collapse to the width (context) dimension: as $T$ grows, bounded logit spreads push attention toward uniform mixing. Our error bound (Appendix D, Eq. 29) formalizes this interaction, showing depth and sequence length jointly determine collapse severity.
>
> > Q2: Results on models trained from scratch are not enough.
>
> To validate beyond NoLiMa, we pre-trained all models from scratch on PG-19 (1K training context) and evaluated perplexity at 2K/4K extrapolation. Due to the rebuttal timeline, models were not trained to full convergence; all architectures were trained for the same number of epochs to ensure a fair comparison of architectural differences.
>
> | Method | Params | PPL @2K ↓ | PPL @4K ↓ | Avg PPL ↓ |
> |---|---|---|---|---|
> | Transformer (Baseline) | 109.8M | 179.86 | 223.20 | 201.53 |
> | Diff Transformer | 109.8M | 97.42 | 220.19 | 158.81 |
> | Transformer + Gated Attn | 113.5M | 77.59 | 153.98 | 115.78 |
> | **Transformer + TRSP** | **109.8M** | **66.40** | **151.33** | **108.87** |
>
> TRSP achieves the **lowest perplexity at both extrapolation lengths** with nearly no additional parameters, confirming that its spectral benefits extend to standard language modeling beyond synthetic tasks like NoLiMa.
>
> > Q3: TRSP does not meaningfully reduce the performance drop with increased context length.
>
> We agree that all methods show some degradation on RULER as context grows, which is expected given the data scarcity (fine-tuned on only 4K context). However, TRSP demonstrates clear advantages across both experimental settings:
>
> **On RULER (post-training)**, TRSP maintains the **highest absolute accuracy at every extrapolation length** (65.7/62.8/60.4 at 8K/16K/32K). Notably, the raw Llama-3.2-1B was pre-trained with a 128K context window, yet standard LoRA fine-tuning on 4K data destroys this native prior—at 32K, LoRA (57.5%) falls below the unmodified raw model (59.6%). TRSP is the only fine-tuned method that consistently outperforms the raw model, showing that its topological regularization acts as a structural safeguard against **catastrophic forgetting** of length generalization during short-context fine-tuning.
>
> **On NoLiMa (pre-training)**, the advantage is far more dramatic, confirming that its spectral guarantees enable genuine length generalization. With no pre-trained prior to rely on, TRSP retains 83.2% at 8× extrapolation while the standard Transformer collapses to 23.8% and the Diff Transformer degrades to 53.9%—gaps of +59 and +29 absolute points.
>
> > Q4: Insufficient citation of prior work on oversmoothing in Transformers.
>
> Thank you for pointing this out. We will expand Section 2.1 to properly discuss foundational work on oversmoothing in Transformers.
>
> ### Conclusion
>
> We thank the reviewer for the thorough evaluation. This rebuttal provides (1) the explicit causal link between spectral collapse and task failure, (2) new PG-19 pre-training results on standard language modeling, and (3) clarified length generalization analysis. **We also validated TRSP on Llama-3.1-8B; details are in our response to Reviewer cDFW, Q2.** We hope these additions address your concerns and kindly ask whether you might reconsider the score.
>
> ### References
>
> [1] Velickovic et al. (2025). Softmax is not Enough (for Sharp Size Generalisation). ICML 2025.
> [2] Barbero et al. (2024). Transformers need glasses! Information over-squashing in language tasks. NeurIPS 2024.

---

> > ### Author Rebuttal · Reviewer_8x8k · 2026-04-04
> >
> > The rebuttal addressed most of my initial concerns. I think the argument linking empirical performance to collapse could be stronger, but I recognize that it does make sense. Could this same argument be made with the PG-19 results? The ppl results are available, but the collapse metrics are not.

---

> > > ### Author Response · Authors · 2026-04-07
> > >
> > > Thank you for the constructive follow-up. We have computed the spectral metrics for all PG-19 models:
> > >
> > > **PG-19 Spectral Metrics & Perplexity (2K / 4K Extrapolation, 109M scale)**
> > >
> > > | Method | Stable Rank ↑ | Anisotropy ↓ | Flatness ↑ | SPR ↑ | PPL ↓ |
> > > |---|---|---|---|---|---|
> > > | Transformer | 4.21 / 3.88 | .215 / .253 | .800 / .813 | 1.092 / 1.099 | 179.9 / 223.2 |
> > > | Diff Trans. | **5.83 / 5.07** | **.161 / .193** | .801 / .819 | 1.076 / 1.078 | 97.4 / 220.2 |
> > > | Gated Attn | 4.57 / 3.64 | .211 / .250 | .816 / .827 | 1.079 / 1.081 | 77.6 / 154.0 |
> > > | **TRSP (Ours)** | 4.63 / 4.04 | .180 / .234 | **.827 / .847** | **1.092 / 1.100** | **66.4 / 151.3** |
> > >
> > > **How these metrics map to theory.** As defined in Section 3.2 and proven in Appendix D, the inference error is upper-bounded by (Eq. 29):
> > >
> > > $$E \leq \overbrace{\frac{\kappa(A)}{\sqrt{sr(X_l)}}}^{\text{Stability}} \cdot \overbrace{\frac{1}{\sqrt{\pi_{\min}}} \cdot (1-\gamma)^L}^{\text{Dynamics}}$$
> > >
> > > Each of our four empirical metrics directly monitors one term in this bound:
> > >
> > > - **Stable Rank** monitors the effective rank $sr(X_l)$ in the denominator of the Stability term [1,2]. A higher Stable Rank means the feature space retains more independent dimensions, shrinking the Stability term and giving the model greater capacity to encode distinct token identities.
> > > - **Spectral Flatness** monitors the condition number $\kappa(A)$ in the numerator of the Stability term [3]. A flatter singular value spectrum means $\kappa(A)$ is closer to 1, ensuring the operator does not amplify errors during forward propagation. Together with Stable Rank, these two metrics characterize the Stability term.
> > > - **Signal Propagation Rate (SPR)** monitors the spectral gap $\gamma$ in the Dynamics term [4]. A higher SPR means a larger $\gamma$, causing the exponential factor $(1-\gamma)^L$ to decay faster—ensuring that information from distant tokens can reach any position within $L$ layers.
> > > - **Anisotropy** monitors the stationary distribution uniformity $\pi_{\min}$ in the Dynamics term [5]. Lower Anisotropy implies a more uniform distribution over tokens, reducing the $1/\sqrt{\pi_{\min}}$ amplification that converts average error into worst-case pointwise error (Lemma D.3).
> > >
> > > Crucially, these four terms are **multiplicative**: a single collapsed metric can dominate the entire bound regardless of how healthy the others are. This is why "spectral balance" is necessary.
> > >
> > > **Analysis.** The PG-19 data perfectly illustrates this multiplicative interaction:
> > >
> > > - **Diff Transformer** (isolation collapse): Best Rank (5.07) and Anisotropy (0.193) at 4K, yet its PPL barely improves over the baseline (220.2 vs. 223.2). The bottleneck is SPR: at 1.078 (the lowest), its differential mechanism preserves spatial geometry but severs long-range information flow, causing the $(1-\gamma)^L$ factor to dominate.
> > > - **Gated Attention** (feature decay): Reasonable at 2K, but its Rank collapses to 3.64 at 4K (worse than the baseline's 3.88), showing the gate aggressively discards features under longer contexts, inflating the $1/\sqrt{sr}$ factor.
> > > - **Standard Transformer** (homogenization collapse): Healthy SPR (1.100) but low Rank (3.88) and high Anisotropy (0.253)—tokens mix well but collapse into a narrow cone, inflating both the Stability and $\pi_{\min}$ terms.
> > > - **TRSP** (spectral balance): The only method without a critical deficit in any dimension. It improves Rank and Anisotropy over the baseline while maintaining the highest SPR and Flatness. No single factor dominates the error bound, yielding the lowest PPL—consistent with our NoLiMa findings (Figures 4–5).
> > >
> > > ### Conclusion
> > >
> > > These metrics provide the empirical closure you identified: spectral health on PG-19 corroborates the causal link to performance, consistent with NoLiMa and Eq. 29. We hope this, combined with our earlier responses, fully addresses your concerns, and would be grateful if you could consider raising the score.
> > >
> > > ### Reference
> > >
> > > [1] Cohen, M. B., Nelson, J., & Woodruff, D. P. (2015). Optimal approximate matrix product in terms of stable rank. arXiv preprint arXiv:1507.02268.
> > > [2] Roy, O., & Vetterli, M. (2007, September). The effective rank: A measure of effective dimensionality. In 2007 15th European signal processing conference (pp. 606-610). IEEE.
> > > [3] Saxe, A., McClelland, J., & Ganguli, S. (2014). Exact solutions to the nonlinear dynamics of learning in deep linear neural networks. Proceedings of the International Conference on Learning Represenatations 2014.
> > > [4] Gouk, H., Frank, E., Pfahringer, B., & Cree, M. J. (2021). Regularisation of neural networks by enforcing lipschitz continuity. Machine Learning, 110(2), 393-416.
> > > [5] Ethayarajh, K. (2019). How contextual are contextualized word representations? Comparing the geometry of BERT, ELMo, and GPT-2 embeddings. In Proceedings of EMNLP-IJCNLP 2019.

---

### Official Review · Reviewer_LofB · 2026-03-14

**Soundness:** 3
**Presentation:** 2
**Significance:** 3
**Originality:** 3
**Overall Recommendation:** 4
**Confidence:** 2

**Summary:**

The paper studies representation collapse in long-context LLMs from a spectral perspective, identifying two pathological extremes: homogenization collapse and isolation collapse. To balance information capacity and mixing efficiency, it proposes TRSP, a lightweight side-path with a hierarchical triangular-box operator and a length-aware gate, showing improved general performance and much stronger length extrapolation on RULER and NoLiMa.

**Compliance With Llm Reviewing Policy:**

Affirmed.

**Key Questions For Authors:**

No.

**Limitations:**

Yes.

**Strengths And Weaknesses:**

Strengths

1. The paper provides a unified view of long-context representation collapse by framing phenomena such as attention sinks, over-mixing, and local attention under a spectral trade-off between effective rank and spectral gap. This perspective is conceptually interesting and helps connect several previously observed issues in transformers.

2. The proposed TRSP module is implemented as a side-path that can be integrated into existing transformer architectures with negligible parameter overhead. Its plug-in nature and linear complexity make it potentially practical for real-world deployment.

Comments

1. Since I am not an expert in this fine-grained area, I am curious about how the proposed method performs on chain-of-thought (CoT) extension and reasoning tasks, where long-range dependency and reasoning depth are particularly important. It would be helpful if the authors could provide additional evaluations in such settings.

2. It would also strengthen the paper to include scaling experiments (e.g., performance under increasing model size or context length). Demonstrating whether the proposed method maintains its benefits under larger-scale settings or weak scaling scenarios would help clarify its practical impact.

---

> ### Author Rebuttal · Authors · 2026-03-31
>
> We sincerely thank the reviewer for the constructive feedback and for recognizing the practical plug-and-play design of TRSP. We are encouraged that all three reviewers consistently acknowledged the value of our unified spectral perspective and the structural efficiency of our approach.
>
> ### Paper Strength
>
> - Unified spectral framework with formal guarantees: We connect previously scattered phenomena (attention sinks, over-mixing, and local attention) under a single spectral trade-off between effective rank and spectral gap, with rigorous mathematical proofs for bounded spectral gap (Theorem C.3) and high effective rank (Propositions 1–2) of the composite operator (Appendix C–D).
> - Efficiency: TRSP introduces only ~50 learnable parameters with linear O(T) complexity, functioning as a non-invasive side-path that preserves pre-trained attention weights.
> - Comprehensive evaluation with strong extrapolation: We validate across general capabilities (MMLU, HellaSwag) and long-context extrapolation in both post-training (RULER) and pre-training (NoLiMa) settings.
>
> We address your specific suggestions with new experiments below.
>
> > Q1: Performance on CoT reasoning tasks with long-range dependency.
>
> To directly address this, we applied TRSP to the larger **Llama-3.1-8B** model fine-tuned on **GSM8K**, a math reasoning benchmark requiring multi-step chain-of-thought:
>
> | Method | GSM8K Eval. Accuracy (%) |
> |---|---|
> | Raw | 10.6 |
> | LoRA | 41.5 |
> | LoRA + Gated Attention | 37.4 |
> | **LoRA + TRSP (Ours)** | **51.1** |
>
> TRSP effectively preserves and enhances CoT reasoning capabilities, outperforming LoRA by **+9.6 points** and Gated Attention by **+13.7 points**. CoT tasks inherently require the model to maintain a precise, uninterrupted logical chain over extended contexts. By mathematically preventing both token homogenization (which blurs distinct logical steps) and context isolation (which severs long-range dependencies), TRSP acts as a structural safeguard. The results above demonstrate that TRSP safely preserves the model's native reasoning depth and yields solid improvements over standard fine-tuning.
>
> > Q2: Scaling experiments under increasing model size or context length.
>
> **Model scaling.** The GSM8K experiment above validates TRSP at 8B scale—8x larger than our main experiments—confirming that its structural benefits transfer effectively to larger models. Notably, Gated Attention again underperforms standard LoRA in the post-training setting, consistent with our 1B findings (Table 1).
>
> **Context scaling.** Our experiments explicitly test this across both settings. On RULER (post-training), TRSP maintains the highest accuracy at every extrapolation length from 8K to 32K (Table 2). On NoLiMa (pre-training from scratch), TRSP retains 83.2% at 8× extrapolation while the standard Transformer collapses to 23.8% (Table 3). Crucially, Figures 4–5 reveal the mechanism: as context length grows, all baselines exhibit monotonic degradation in spectral health (declining Stable Rank, rising Anisotropy, etc.), with accuracy dropping in lockstep. TRSP alone preserves healthy spectral metrics throughout, confirming that its length generalization stems directly from structural prevention of representation collapse.
>
> ### Conclusion
>
> We appreciate the reviewer's constructive suggestions, which led us to supplement 8B-scale CoT experiments. We hope these results strengthen confidence in TRSP's practical impact, and would be grateful if you could consider raising the score.

---

> > ### Author Rebuttal · Reviewer_LofB · 2026-04-04
> >
> > Thanks! I will keep my score.

---

### Official Review · Reviewer_cDFW · 2026-03-18

**Soundness:** 3
**Presentation:** 2
**Significance:** 2
**Originality:** 3
**Overall Recommendation:** 4
**Confidence:** 3

**Summary:**

This work addresses long-context representation collapse in transformers through spectral and topological regularization. The main contribution is TRSP, a lightweight side-path combining a hierarchical causal triangular smoothing operator with a length-aware gate, alongside a spectral framework that unifies homogenization collapse and isolation collapse under one lens. Empirically, the method shows modest but consistent gains on general benchmarks and notably strong extrapolation on NoLiMa, though the theoretical guarantees mainly apply to the auxiliary branch and the empirical validation remains somewhat narrow.

**Compliance With Llm Reviewing Policy:**

Affirmed.

**Key Questions For Authors:**

The current experiments are limited to 1B and 109M models. Do the authors have any evidence or analysis, even preliminary, on how TRSP's benefits scale with model size?

**Limitations:**

Yes

**Strengths And Weaknesses:**

Strengths
- The paper's contribution is unifying previously scattered phenomena, such as over-mixing, attention sinks, and the limitations of local attention, under a single spectral analysis framework, revealing that they are two extremes of the same fundamental trade-off between spectral gap and effective rank.
- TRSP introduces only approximately 50 learnable parameters.
- TRSP is added as a parallel side-path without modifying the pre-trained attention weights. T


Drawbacks

- The improvements on general benchmarks are modest
- All experiments are conducted on 1B or 109M-parameter models. At this scale, it is difficult to determine whether the method remains effective on larger models such as 7B or 13B, which is important given that the severity of representation collapse may vary with model scale.
- The theoretical guarantees do not truly extend to the entire network. The authors themselves acknowledge in the Remark on page 5 that the rigorous spectral guarantees apply to the TRSP branch itself, not to the final composite operator I + Attn + TRSP under worst-case conditions. Therefore, the claim that "TRSP guarantees a high effective rank and a bounded spectral gap" is, in a strict sense, more accurately described as holding for the side-path, while serving as an inductive bias rather than a worst-case guarantee for the full network.

---

> ### Author Rebuttal · Authors · 2026-03-31
>
> We sincerely thank the reviewer for the constructive feedback and for recognizing the value of our unified spectral framework, as well as TRSP's lightweight and non-invasive design. We address your specific concerns below:
>
> > Q1: The improvements on general benchmarks are modest.
>
> We respectfully clarify that these modest gains are expected, as general benchmarks (e.g., MMLU) predominantly utilize short contexts ($\le\$ 1K) where baseline models are already well-optimized; TRSP's true architectural advantage lies in preventing catastrophic collapse in long-context scenarios. At short lengths, representation collapse has not yet triggered, so TRSP safely preserves base capabilities with minor gains. Its necessity shines at extended lengths (e.g., 32K on RULER), where TRSP structurally prevents the cliff-edge degradation inherent to standard transformers.
>
> > Q2: Scalability and effectiveness on larger models (e.g., 7B/13B).
>
> Yes, TRSP's benefits effectively scale to larger models. We applied TRSP to Llama-3.1-8B fine-tuned on GSM8K (math reasoning), and observed consistent improvements:
>
> | Method | GSM8K Eval. Accuracy (%) |
> |---|---|
> | Raw | 10.6 |
> | LoRA | 41.5 |
> | LoRA + Gated Attention | 37.4 |
> | **LoRA + TRSP (Ours)** | **51.1** |
>
> TRSP outperforms LoRA by **+9.6 points** and Gated Attention by **+13.7 points** on this 8B model—a larger absolute gain than at 1B scale. Notably, Gated Attention again underperforms standard LoRA in the post-training setting, consistent with our findings in Table 1. These results confirm that TRSP serves as a robust topological regularizer regardless of model scale, with particular benefits on tasks requiring multi-step reasoning where spectral health directly impacts information propagation.
>
> > Q3: Do the spectral guarantees apply to the full composite network or only the TRSP side-path?
>
> We apologize that the overly conservative wording of the Remark on page 5 obscured the stronger results already present in our appendix; we will revise it to avoid this confusion. **We clarify that Appendix C (Lemma C.2, Propositions 1–2) already establishes rigorous worst-case guarantees for the full composite operator $\mathcal{M} = I + A_{\mathrm{attn}} + \lambda T_{\mathrm{tri}}$, not just the TRSP branch.**
>
> **1. Spectral gap: composite guarantee (Lemma C.2).**
> Lemma C.2 is formulated for the composite operator. Since graph Laplacians are positive semi-definite, the algebraic connectivity of a graph union is at least the sum of its components' gaps (Weyl's monotonicity [2]). Even if attention fully collapses, TRSP's constant gap (Theorem C.3) provides a deterministic, worst-case lower bound for the entire network.
>
> **2. Effective rank: non-negativity eliminates cross-term risk (Propositions 1 & 2).**
> The effective rank could only be harmed if Frobenius cross-terms were negative. However, the identity, Softmax attention, and the triangular kernel are all element-wise non-negative, so every cross-term is constructive. Proposition 1 shows the composite energy grows linearly with sequence length; Proposition 2 shows that even under worst-case sink collapse, the composite effective rank remains strictly above 1.
>
> **3. Destructive interference is structurally impossible.**
> The only failure mode would be exact cancellation between attention and TRSP ($A_{\mathrm{attn}} \approx -\lambda T_{\mathrm{tri}}$), requiring negative attention weights. Since Softmax outputs are strictly positive, this is architecturally impossible. Moreover, the structural incoherence between sparse attention sinks and TRSP's smooth circulant basis [1, 4] ensures negligible cross-interference—consistent with our empirical spectral metrics (Figure 4).
>
> We will update the Remark to explicitly state these composite guarantees.
>
> ### Conclusion
>
> We deeply appreciate your rigorous evaluation, which has helped us significantly tighten our theoretical presentation. We hope that the newly added 8B-scale experiments and the detailed justification of full-network spectral guarantees have thoroughly addressed your core concerns. We kindly ask whether you might reconsider the score in light of these additions, and remain at your disposal for any further clarifications.
>
> ### References
>
> [1] D. L. Donoho and X. Huo. 2006. Uncertainty principles and ideal atomic decomposition. IEEE Trans. Inf. Theor. 47, 7 (November 2001), 2845–2862.
> [2] Stewart, G. W., & Sun, J.-G. (1990). *Matrix perturbation theory*. Academic Press.
> [3] Davis, C., & Kahan, W. M. (1970). The rotation of eigenvectors by a perturbation. III. SIAM Journal on Numerical Analysis, 7(1), 1-46.
> [4] Candès, E.J., Recht, B. Exact Matrix Completion via Convex Optimization. Found Comput Math 9, 717–772 (2009).
> [5] Vershynin, R. (2018). High-Dimensional Probability. Cambridge University Press.

---

> > ### Author Rebuttal · Reviewer_cDFW · 2026-04-04
> >
> > Thank you for the detailed rebuttal. I appreciate the clarification on the theoretical guarantees and the additional larger-scale evidence.
> >
> > Overall, my main concerns have been adequately addressed. These clarifications strengthen my confidence in the paper, but they do not materially change my overall assessment of its significance, so I keep my original score.

---

### Decision · Program_Chairs · 2026-04-30

**Decision:**

Accept (regular)

**Comment:**

This paper studies representation collapse in long-context LLMs through a spectral lens, unifying homogenization collapse and isolation collapse as two extremes of a trade-off between mixing efficiency and information capacity. This study's major theme pertains to understanding and mitigating this collapse via spectral/topological regularization, and Overall, this submission's significant contribution consists of proposing TRSP, a lightweight side-path that combines a hierarchical triangular-box operator with a length-aware gate to improve spectral balance without modifying the core attention mechanism.

I recommend acceptance. The paper offers a clear conceptual contribution by reframing several previously separate long-context failure modes under one coherent spectral framework, and it backs this up with a practical intervention that is extremely lightweight and easy to integrate. Empirically, TRSP gives modest but consistent gains on general benchmarks and much stronger improvements on long-context extrapolation, where representation collapse is most relevant.

The main concerns were about scale and the scope of the theoretical guarantees, and the rebuttal addressed these points well by clarifying the full-network guarantee claims and adding larger-scale evidence, including 8B-model GSM8K results and further long-context justification. Reviewers indicated that these concerns were resolved while maintaining positive recommendations.

Overall, this is a solid ICML paper with an interesting unifying perspective, a practical method, and convincing long-context results, so I recommend acceptance.